# Localizing strain via micro-cage structure for stretchable pressure sensor arrays with ultralow spatial crosstalk

Yufei Zhang [1,2,6], Qiuchun Lu[1,3,6], Jiang He[1,4,6], Zhihao Huo[1,2], Runhui Zhou[1,2], Xun Han[1], Mengmeng Jia[1,2], Caofeng Pan [1,2,3] ✉, Zhong Lin Wang [1,2,5] ✉ & Junyi Zhai [1,2,3] ✉

Tactile sensors with high spatial resolution are crucial to manufacture large scale flexible electronics, and low crosstalk sensor array combined with advanced data analysis is beneficial to improve detection accuracy. Here, we demonstrated the photo-reticulated strain localization films (prslPDMS) to prepare the ultralow crosstalk sensor array, which form a micro-cage structure to reduce the pixel deformation overflow by 90.3% compared to that of conventional flexible electronics. It is worth noting that prslPDMS acts as an adhesion layer and provide spacer for pressure sensing. Hence, the sensor achieves the sufficient pressure resolution to detect 1 g weight even in bending condition, and it could monitor human pulse under different states or analyze the grasping postures. Experiments show that the sensor array acquires clear pressure imaging and ultralow crosstalk (33.41 dB) without complicated data processing, indicating that it has a broad application prospect in precise tactile detection.

Development of large-scale and high-density flexible sensor arrays could provide better human–machine interaction[1–8], and high-precision tactile perception could be achieved by analyzing the measured data[9,10] or isolating different crosstalk[11–16]. At present, signal crosstalk in sensor arrays is mainly divided into two types. One is the crosstalk among electrical signals, such as leakage, breakdown, or the external electromagnetic interference, etc. This phenomenon could be alleviated by using inductance and capacitance, or solved by the signal processing algorithm. The other type is so-called mechanical crosstalk, which often occurs in flexible electronics. When a pixel receives an external pressure, its deformation will inevitably spread to surrounding regions, so the adjacent pixels will also respond. Crosstalk may be used to localize the external stimulus, but it usually requires further analysis of the measured signal for higher detection accuracy.

Recently, intensive endeavors have been devoted to fabricate the novel materials for the preparation of high-performance flexible tactile sensors, such as metal nanofibers[17–22], carbon nanomaterials[23–32], conducting conjugated polymers[33–38], and ionic conductors[39–41]. Typically, Bao et al.[42,43] proposed the sensor arrays based on the capacitive principle, which possess the excellent transparency and sensitivity. However, due to the lack of effective strategy to isolate mechanical crosstalk, the stressed pixel has a significant impact on the adjacent pixels. Especially for multi-pixel perception, even the exact force point cannot be distinguished. Moreover, combining pressure-sensitive rubber (PSR) with the transistors to fabricate tactile sensor is also a common method[44–46]. When the external pressure changes the conductance of the PSR, the source-drain current also changes in accordance with pressure, which will sufficiently improve the sensor

[1]CAS Center for Excellence in Nanoscience, Beijing Key Laboratory of Micro-nano Energy and Sensor, Beijing Institute of Nanoenergy and Nanosystems, Chinese Academy of Sciences, Beijing 100083, P. R. China. [2]School of Nanoscience and Technology, University of Chinese Academy of Sciences, Beijing 100049, P. R. China. [3]Center on Nano-Energy Research, School of Physical Science & Technology, Guangxi University, Nanning 530004, China. [4]Siyuan Laboratory, Guangdong Provincial Engineering Technology Research Center of Vacuum Coating Technologies and New Energy Materials, Department of Physics, Jinan University, Guangzhou, Guangdong 510632, China. [5]School of Materials Science and Engineering, Georgia Institute of Technology, Atlanta, GA, USA. [6]These authors contributed equally: Yufei Zhang, Qiuchun Lu, Jiang He. ✉e-mail: cfpan@binn.cas.cn; zhong.wang@mse.gatech.edu; jyzhai@binn.cas.cn

sensitivity. Nevertheless, in order to make the source electrode contact with PSR layer, the entire device should be designed with vias, which will undoubtedly cause crosstalk between different pixels. To meet more sophisticated application scenarios, tactile sensors are still developed in the direction of being lighter, thinner and able to adhere on irregular surfaces[47–52]. An ultra-light and imperceptible plastic electronics[53] has been demonstrated for pressure imaging and could be applied on more complex surfaces such as oral cavity, these higher requirements also aggravate the mechanical crosstalk in flexible electronics. Furthermore, Oddo[10] et al. reported a biomimetic skin based on the photonic fiber Bragg grating transducers with overlapping receptive fields, and the force and localization predictions could be realized by the convolution neural deep learning algorithm. This technology combined with an efficient calculation power system will show broad application prospects in the field of collaborative robots. As discussed above, proposing an effective strategy to optimize the sensor structure to achieve direct measurement of external signals can not only greatly reduce in-depth analysis of data, but also effectively achieve precise tactile detection in high spatial resolution sensors.

Here, we demonstrated the photo-reticulated strain localization films (prslPDMS), which is suitable for the preparation of ultralow crosstalk sensor due to its micro-cage structure. Compared with the flexible electronics on PET substrate, prslPDMS could reduce the pixel deformation overflow by 90.3% in sensor arrays. The ultrathin pressure sensor was assembled by stacking the interdigital Ag NFs, prslPDMS, and micropatterned graphene layer, and its optimal detection range could be tuned by controlling the thickness of prslPDMS layer. It is worth noting that the ultrathin sensors still acquire the high-pressure resolution to detect the 1 g (~150 Pa) weight even in bending condition, so it could measure human pulse in different states (drinking, exercise, and rest) or analyze the grasping postures. Experimental data show that the sensor arrays have ultralow crosstalk (33.41 dB) and enable clear pressure imaging, which could control the model movement by using a customized program. Moreover, according to the mechanical simulation, in case of the spacer length and pixel length being 1 μm and 5 μm, respectively, the crosstalk isolation is 22.43 dB and the pixel spatial resolution could exceed 4000 ppi. We predict that the proposed device has a broad application prospect in the field of wearable electronics, soft robotics, and human–machine interaction.

## Results

### Design concept of micro-cage structure based on prslPDMS layer

Human sense of touch deals with spatiotemporal perception under external stimuli through a large number of receptors (Fig. 1a). The relevant information reaches the spinal cord through multiple nerves and is transmitted to the central nervous system via two main pathways for higher-level processing and interpretation: spinothalamic and dorsal-column-medial-lemniscal. The latter could quickly convey pressure/vibration information to the brain for precise tactile detection. Taking fingertips as an example, there are many mechanoreceptors embedded in skin at different depths, which are mainly divided into four categories: slow-adapting receptors responding to static pressures (SA-I and SA-II) and fast-adapting receptors responding to dynamic forces or vibration (FA-I and FA-II). These mechanoreceptors show the different receptive fields, and they work together to achieve the precise tactile perception. Our pressure sensor adopts another strategy to achieve the above effect, which is to separate the receptive fields by introducing a dielectric layer, thereby preparing the tactile sensor with ultralow crosstalk. As shown in Fig. 1b, doping benzophenone into PDMS will inhibit its cross-linking when exposed to UV light, thus forming the photo-reticulated PDMS (Supplementary Fig. 1). The micro-cage structure is formed after encapsulating with another layer of electrodes, and pressure sensor could be prepared within the cage. Besides, the boundary of the micro-cage composed by

photo-reticulated PDMS could separate different sensors and prevent unnecessary deformation diffusion, showing the effect of strain local confinement, so it is called photo-reticulated strain localization PDMS (prslPDMS). A simple two-dimensional model is proposed to analyze the deformation of micro-cage structure under external pressure (Fig. 1c). It could be found that the entire model has four key parameters, which are pixel length ($l_p$), spacer length ($l_s$), and thickness ($t_s$), and external displacement ($D$). When a pixel is subjected to external pressure, both the top electrode and prslPDMS spacer will deform, thereby gradually expanding to adjacent pixel, resulting in mechanical crosstalk. Deformation simulation analysis is performed on PDMS model with prslPDMS spacer and the model only with PET (Fig. 1d). The results demonstrate that the former realizes the small elongation strain due to the strain confinement of prslPDMS spacer, but PET substrate with high toughness and non-stretchable characteristics shows the large deformation. Moreover, a coordinate system is established on the right boundary of stressed pixel to quantitative analyze the elongation strain along the x-axis and y-axis (Fig. 1e). The displacement along the y-axis declines slowly for the PET model, and its average displacement in adjacent pixel is 6.13 μm. However, the deformation of prslPDMS spacer model decreases rapidly in the spacer layer region, and the average displacement of adjacent pixel is only 0.56 μm, which decreases by 90.3% compared with PET substrate (the maximum deformation of stressed pixel). Figure 1f describes the crosstalk isolation versus the different ratios of spacer length to pixel length, and its value reaches 25.03 dB with the ratio of 0.5, which can be considered as better mechanical crosstalk isolation (More detailed analysis could refer to the Supplementary Fig. 10).

### Structure of pressure sensor array

Figure 2a shows the exploded view of the stretchable ultralow crosstalk sensor array. The device mainly consists of three parts: the patterned Ag NFs interdigital electrodes, the patterned prslPDMS layer, and graphene attached to the PDMS with pyramid microstructures. The main challenge in this work is to prepare the patterned dielectric films (prslPDMS), which can not only separate adjacent pixels to form micro-cage structure, but also show the function of adhesion and support for packaging devices. SEM images of prslPDMS layer with different resolutions are shown in Fig. 2b, with a precision up to 100 μm. Then the patterning effect on different substrates was also verified, such as glass (horse) and silicon (office logo), indicating its excellent adaptability. In addition, the stretchability and transmittance of prslPDMS layer were further demonstrated in Fig. 2c. The results show the prslPDMS layer possesses the similar stretchability (~5.08 MPa) and transmittance (91.85% in visible light) with PDMS, allowing the preparation of transparent stretchable devices. For more detailed fabrication process, please refer to the "Methods", Supplementary Note 1–3, and Supplementary Movie 1. Since the high precision transparent stretchable prslPDMS film could achieve strain local confinement, it can be used to prepare multilayer devices and improve the mechanical stability, which could provide a solid foundation for more sophisticated electronics (Fig. 2d). Figures 2e and 2f show the 6 × 6 sensor array is well attached on the palm no matter in the flat or curly state. The insert figure (Fig. 2e) exhibits the single sensor on fingertip (2.0 × 2.0 mm²) with a transmittance of 50.36%, and the enlarged cross-section SEM image (Fig. 2f) shows the thickness of multilayer stacked structure is only 60.59 μm.

### Characterization of pressure sensing properties

The pressure sensing performance of the sensor is investigated by using a self-made measurement system, as shown in Fig. 3a. Basically, the resistance of the sensor mainly depends on the contact resistance between the two electrodes, which will decrease rapidly with the external pressure. The sensitivity ($S$) in this study could be obtained by the formula $S = \partial(\Delta I/I_0)/\partial P$, the ratio of the relative change in current

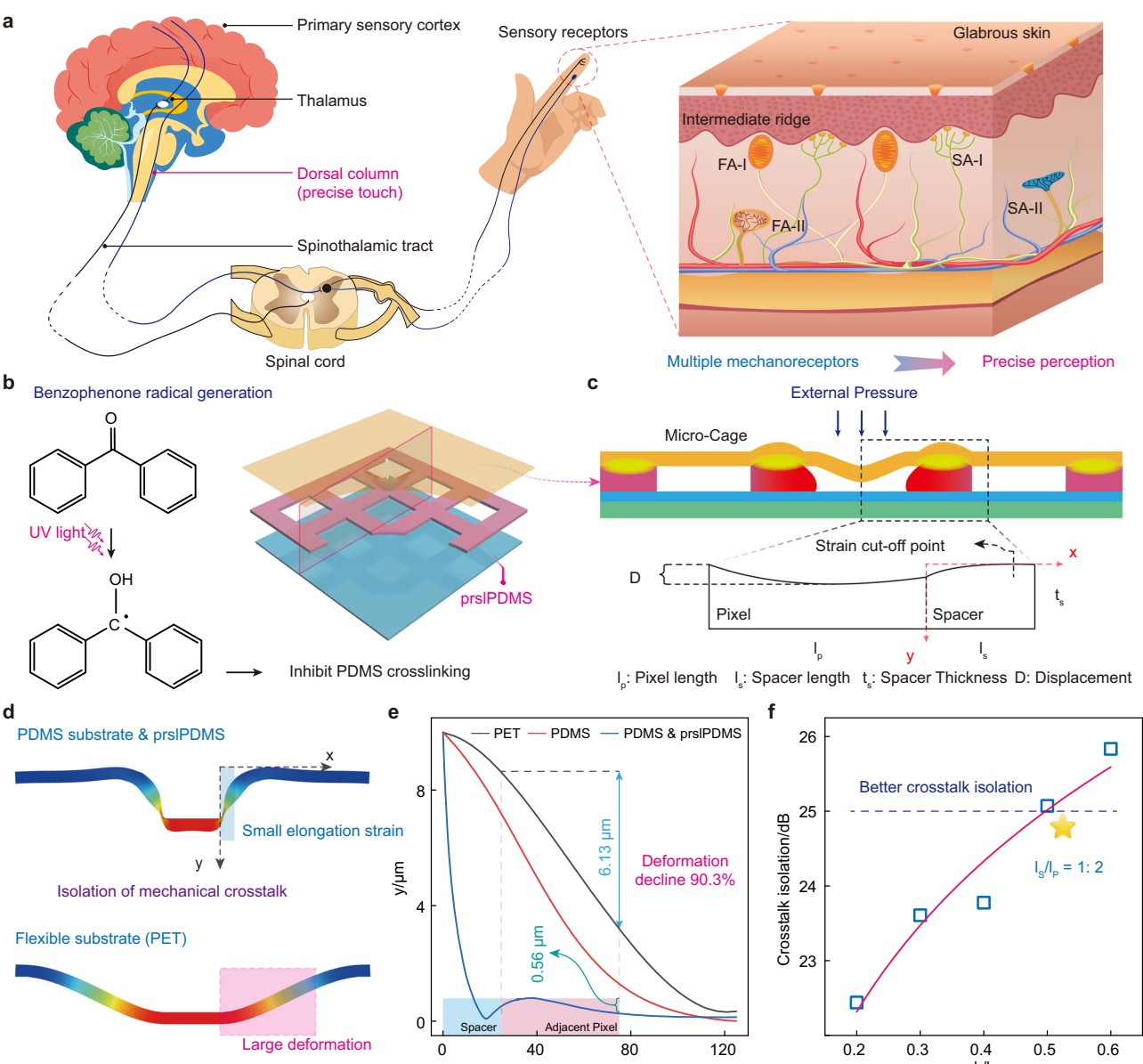

**Fig. 1 | Principle of ultralow crosstalk sensor with micro-cage structure based on prslPDMS layer. a** Schematic illustration of multiple mechanoreceptors in glabrous skin for accurate tactile perception. **b** Proposed chemistry of benzophenone inhibiting PDMS cross-linking under UV light to prepare prslPDMS layer. **c** Simple geometric analysis of micro-cage structure formed by prslPDMS under external pressure. **d** Two-dimensional deformation simulation analysis of the PDMS & prslPDMS and PET under external pressure, and the micro-cage structure of the former enables the small elongation strain compared with the latter. **e** Quantitative analysis of displacement variation along the x-axis for different models, and the displacement of adjacent pixel using prslPDMS layer decrease by 90.3% compared with PET. **f** Crosstalk isolation versus the ratio of spacer length to pixel length, the better isolation effect could be achieved when its value is greater than 1:2.

to the applied pressure. Extensive experimental data (Fig. 3b) show that the sensitivity curve could be divided into three regions, the low-pressure region, the linear region, and the saturation region. These regions correspond to the sensor subjected to different pressures, and the simulations results are shown below. Only the pyramidal tip has the stress concentration with low pressure, so its contact area is relatively small. Then the pyramid and upper substrate gradually contact with the increasing pressure, resulting in the greater stress concentration and larger deformation of the whole electrode. Eventually, the ultra-thin upper substrate deforms thoroughly, making the multilayer structure in a close contact (More simulation results could refer to Supplementary Note 4), and the SEM images in different states are depicted at the bottom. Figure 3c illustrates the relative change in current ($\Delta I/I_0$) versus applied pressure for a series of sensor with different interdigital silver nanofibers (Ag NFs) electrodes. Denser

interdigital electrodes show the larger current variation, and the distinction of three regions becomes more obvious, which can be attributed to the generation of more contact sites, resulting in the high sensitivity sensor. The effect of prslPDMS spacer thickness on sensor performance is shown in Fig. 3d. As the spacer thickness increases, there will be larger air gap in the sensor, which means the more pressure could make the sensor responsive. Therefore, although the sensitivity curves shift to the right, they have similar trend and the average sensitivity of the three regions are 0.40, 1.88, and 0.18 kPa$^{-1}$, respectively. Furthermore, Fig. 3e presents the properties with the different pyramids ranging from 10 μm to 60 μm. The sensor with the pyramidal size of 10 μm has a sensitivity of 18.94 kPa$^{-1}$, but its linear detection range is below 20 kPa. Detection range is significantly improved with the greater pyramid size, but its sensitivity drops rapidly to 0.015 kPa$^{-1}$ at the pyramid size of 60 μm. The sensors with

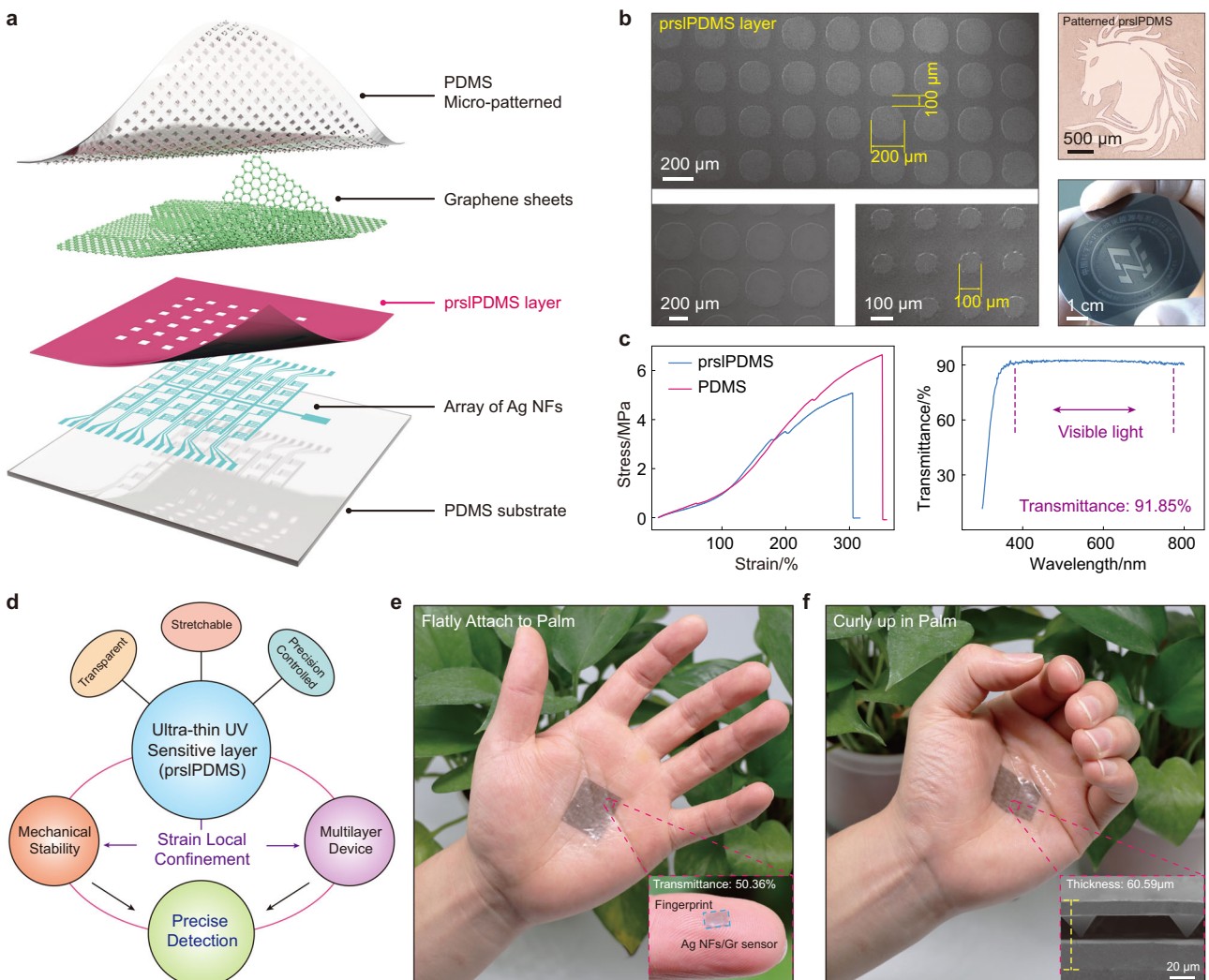

**Fig. 2 | Structure and design concept of the pressure sensor. a** Schematic illustration of the structure of pressure sensor arrays. **b** SEM images of prslPDMS layer with different resolutions, which could reach 100 μm; the right photographs demonstrate the patterns on various substrates (top: glass, bottom: silicon wafer), indicating its good adaptability. **c** Tensile properties (left) and UV-visible spectra (right) of prslPDMS exhibit the excellent stretchability and transparency. **d** Schematic showing the design concept for the ultralow crosstalk sensor, where the prslPDMS layer perform the effect of strain local confinement for mechanical stability as well as multilayer device. Optical photographs of sensor arrays attached on palm no matter in the flat (**e**) or curly (**f**) state, which achieve the transparency of 50.36% and thickness of 60.59 μm.

the smaller pyramids will be in close contact under less pressure, so they realize the higher sensitivity, but their detection range is relatively narrow. Since the larger pyramid may destroy the integrity of the electrodes, the detection range is substantially improved at the expense of the sensitivity, so it is necessary to design an appropriate pyramid size to satisfy the practical application.

Further tests were carried out to demonstrate the stretchability of the pressure sensor, and it could be found that the sensitivity gradually decreases from 2.23 kPa$^{-1}$ to 0.68 kPa$^{-1}$ with the deformation increasing to 50% (linear region). However, its performance return to the initial state after releasing the deformation, suggesting the excellent resilience (Fig. 3f). Figure 3g illustrates the effect of bending on the sensor performance, both tensile and compressive bending could cause the current variation, which changes slowly at the large bending radius, but increases rapidly when the radius reduces below 6.94 mm. The sensor could distinguish the tiny weights of 1 g, 2 g, and 5 g at the bending radius of about 5 mm, suggesting the good pressure resolution. This could be ascribed to the adhesion and support of prslPDMS layer, which not only ensures the stacked structure won't separate, but also provides the spacer

for pressure sensing even in the bending state (More detailed analysis can refer to Supplementary Fig. 16). The durability of the sensor was detected by a repetitive contact test for more than 5000 cycles and the relative change in current almost unchanged, exhibiting an excellent stability (Fig. 3h). Experimental data also demonstrate that thicker spacer layer slightly increases the response time and recovery time, and the sensor has the rapid response time (0.284 s) and recovery time (0.102 s) with the spacer layer thickness of 15 μm. As discussed above, the sensor with denser interdigital electrodes, thinner prslPDMS layer and 30 μm pyramidal microstructure exhibits the sensitivity of 2.53 kPa$^{-1}$ with the pressure range from about 20 to 250 kPa, so the subsequent investigation will be based on this structure to obtain the better performance.

**Human pulse detection and grasping posture analysis**

Figure 4a shows a schematic diagram of the sensor attached to various arteries in human body, including carotid, wrist, and ankle. Briefly, blood pressure is usually expressed as two components, diastolic and systolic pressure, which could be characterized by the

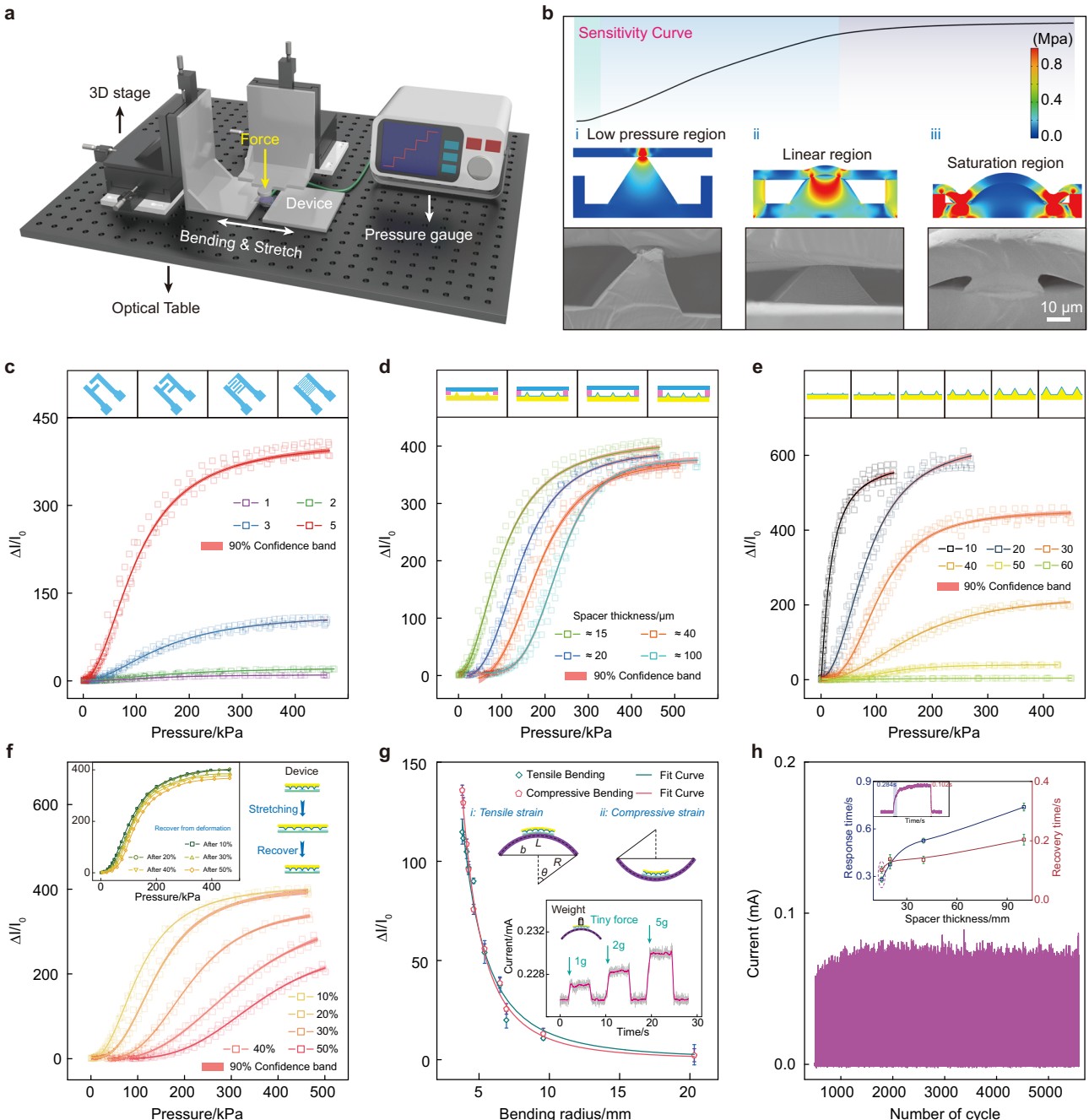

**Fig. 3 | Pressure sensing performance of the ultrathin sensor. a** Schematic diagram of the experimental setup for pressure sensing. **b** Sensitivity curve of the sensor (top); Simulation of stress distribution of the prslPDMS layer and microstructured PDMS under external pressure (middle); Cross-sectional SEM images of the sensor under different pressures (bottom). Relative change in current ($\Delta I/I_0$) of the sensors with different interdigital Ag NFs electrodes (**c**), spacer layer thickness (**d**), and pyramid sizes (**e**). Relative change in current ($\Delta I/I_0$) of the sensor with different strains (**f**) and bending radii (**g**). Inset: sensor response after recovery from deformations (left) and to tiny forces even in the bending state (right). **h** Stability measurement of the pressure sensor, and the cycle period is over 5000 cycles. Inset: the response performance of the sensor with different spacer thickness.

current variation of the ultrathin sensor. Three different pressures (20 kPa, 10 kPa, 5 kPa) are sequentially applied to the radial artery with the ultrathin sensor via the traditional blood pressure cuff (Fig. 4b). The sensor exhibits the larger base current with the increasing pressure, and it also reveals a stable current fluctuation which contains the pulse information of the human body (Supplementary Movie 2). Further analysis of the magnified pulse waveform is displayed in Fig. 4c. The typical characteristic pulse waveform was observed with three clearly distinguishable peaks even under different pressure, systolic peak ($P_S$), point of inflection ($P_i$), and dicrotic

wave ($P_D$). Two commonly used parameters could be obtained: augmentation index $AIx(\%) = P_i/PP$, and the reflection index $RI = h/\Delta t$, where $h$ is the height of subject, $\Delta t$ is the time interval between $P_S$ and $P_D$, and $PP$ is the absolute magnitude of the pulse waveform. Figure 4d proves the human pulse detection of ultrathin sensors in a variety of application scenarios, including drinking, exercise, and rest. A 25-year-old male has a normal pulse rate of 1.267 Hz, and it becomes 1.533 Hz after proper drinking. Then it further rises to 1.724 Hz after a short period of exercise, finally returns to 1.533 Hz through the adequate rest. The corresponding $AIx$ and $RI$ are

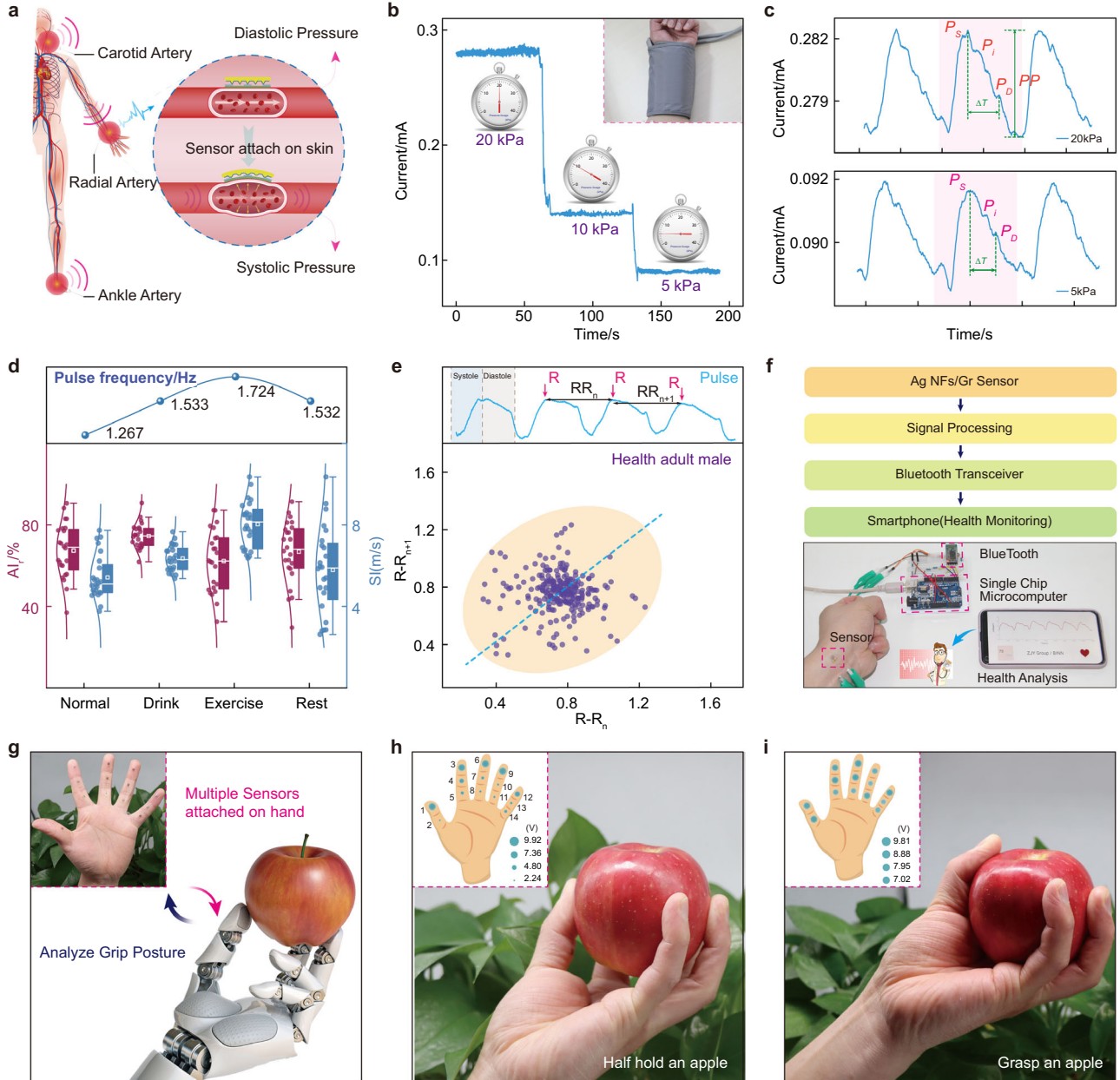

**Fig. 4 | Real-time detection of human pulse and analysis of grasping posture.** **a** Schematic diagram of sensors attached to the various arteries and the principle to detect the systolic and diastolic pressure. **b** Pulse signal detected with different external pressures. Inset: using a blood pressure cuff to apply pressure. **c** Extracted interpreted pulse waveforms under the pressure of 20 and 5 kPa. **d** Augmentation index (*AIx*), reflection index (*RI*), and frequency of the pulse waveforms in different states, including drinking, exercise, and rest. Top: Pulse frequency variation

shown below, and more detailed analysis could refer to Supplementary Note 6. Figure 4e shows the ellipse-shaped Poincare plot of a healthy adult male, which could exhibit the heart rate variability and predict cardiovascular disease. Based on the above experimental results, the complete wireless pulse monitoring system is shown in Fig. 4f. The pulse signal detected by the ultrathin sensor will be processed by the single-chip microcomputer and sent to the smart phone through Bluetooth, then the measured pulse waveform and heart rate could be observed on the App interface. Multiple sensors are attached on the finger to analyze the grasping posture, as shown in Fig. 4g. Fourteen sensors are sticked on the phalanges of the palm, and the fine copper wires with a diameter about 30 μm was used to

obtained by Fourier transform. **e** Poincare plot of a 25-year-old male for a long-term pulse detection of 200 s. **f** Real-time pulse wave shown on a smart phone via App interface for human health monitoring. **g** Multiple sensors attached to the front of the finger to detect the different grasping postures. The sensors at fingertip have large voltage with a half hold apple (**h**), and all sensors have a higher voltage when the apple is fully grasped (**i**).

lead out the electrodes (Supplementary Fig. 18). In order to facilitate multichannel data acquisition, each sensor is connected in series with an external resistor. By detecting the voltage of resistors, the current variation in the circuit could be calculated, thereby reflecting the pressure on the sensor. More detailed analysis could refer to the Supplementary Fig. 19. Figure 4h reveals the data when an apple is half held, the sensors at the fingertips show the larger voltage, but all sensors have the high voltage with the apple fully grasped (Fig. 4i, Supplementary Fig. 20, and Supplementary Movie 3). These results are consistent with the actual situation, so our ultrathin sensors possess the excellent performance in both single sensor precise measurement and collaborative detection of multiple devices.

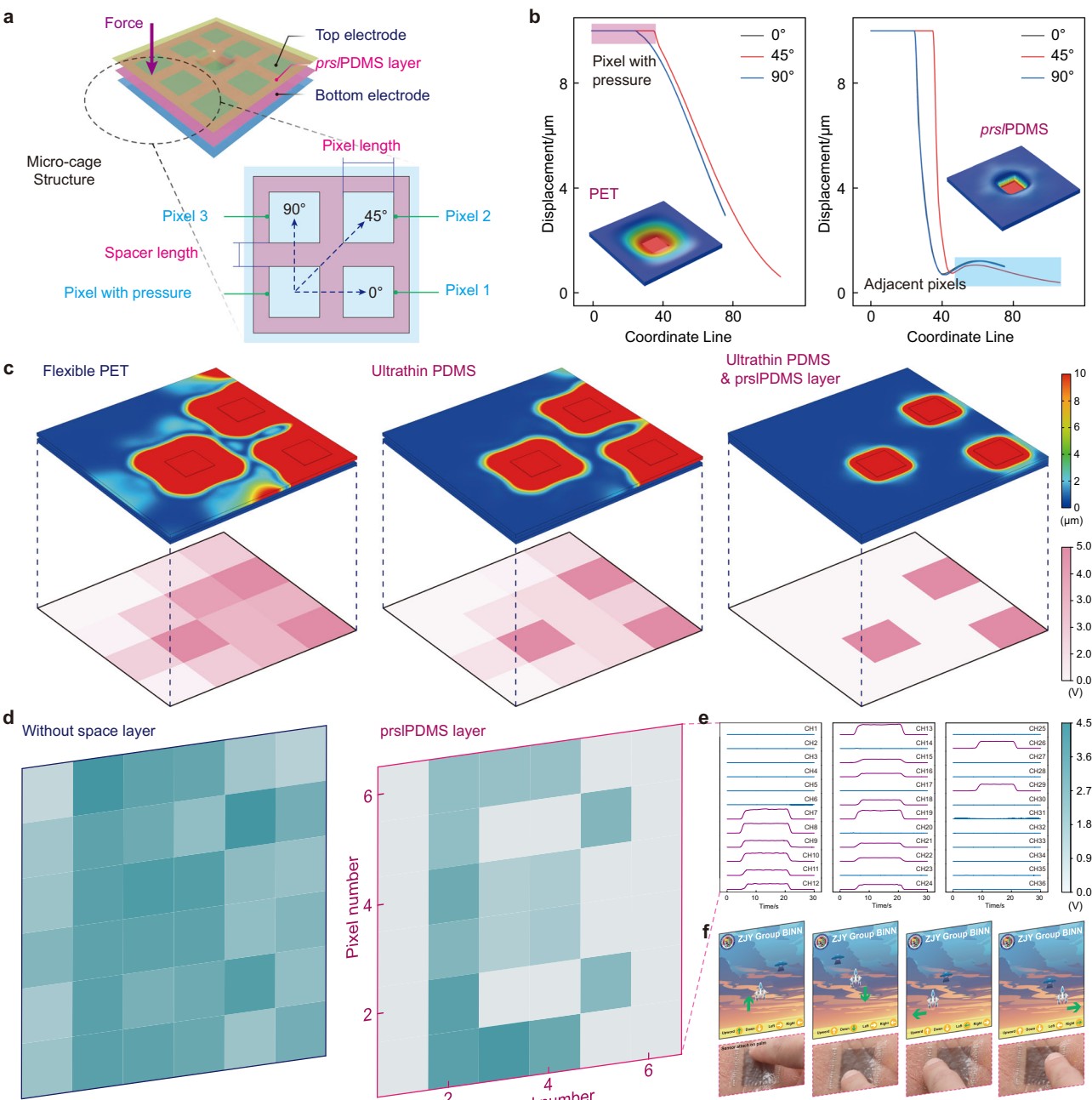

**Fig. 5 | Measurement and analysis of ultralow crosstalk sensor arrays. a** Three-dimensional analysis of micro-cage structure formed by prslPDMS layer to isolate mechanical crosstalk, and the stressed pixels will affect the adjacent pixels from three directions (0°, 45°, 90°). **b** Displacement of PET and prslPDMS spacer model in different directions, and the model with spacer layer has almost no additional deformation to adjacent pixels. **c** Crosstalk simulations and corresponding voltage variation of sensors based on PET, PDMS and PDMS & prslPDMS layer, it could be found that the sensor arrays using the prslPDMS layer has less mechanical crosstalk and a clear pressure imaging. **d** A 6 × 6 sensor array used for pressure detection, and the imaging of the letter "B" could be seen clearly for the device with the prslPDMS layer. **e** Multichannel voltage variation of sensor array to detect the letter "B". **f** Demo of application scenarios. The pixels of the sensor array are divided into four regions, which could be used to control the model movement in the game.

## Ultralow crosstalk tactile sensor array

Figure 5a shows the micro-cage structure formed by prslPDMS layer to isolate mechanical crosstalk in three-dimensional model. The pixel with pressure will deform adjacent pixels from three directions (0°, 45°, 90°). The simulated displacement data show the strain localization occurs after using the prslPDMS layer, which is consistent with the previous conclusion. It is also found that the deformation in the 0° direction is equivalent to the 90° direction, and the pixel in the 45° direction is farther from the stressed pixel, so its displacement is smaller than the other two pixels (Fig. 5b). Further multi-pixel compression analysis was carried out in sensor arrays based on PET, PDMS, and PDMS substrate with prslPDMS layer, and the stressed pixels are at the center, boundary or corner. Similarly, the large Poisson's ratio of PET will lead to its lack of stretchability, so when three pixels are under external forces at the same time, it will inevitably cause the surrounding pressure. The deformation spillover to adjacent pixel could be alleviated by using the stretchable materials, such as PDMS, but it still cannot be eliminated well. The cage-structured sensor based on prslPDMS layer could effectively avoid the diffusion of strain, and clear multi-pixel stimulation could be obtained through

**Table 1 | Performance comparison with state-of-the-art pressure sensor**

| Sensing mechanisms | Sensitivity/kPa$^{-1}$ | Detection range/kPa | Stretchability/% | Crosstalk isolation/dB | Studies |
|---|---|---|---|---|---|
| Contact resistance | 18.94 (10 μm) | <40 | ~50 | ~33.41 | This work |
| | 4.04 (30 μm) | <200 | | | |
| Contact resistance | 4.88 | 0.37–8 | ~40 | – | Ref. [61] |
| Contact resistance | 0.7 | <25 | <3 | Low | Ref. [28] |
| Contact resistance | ~15.1 | 0.2–59 | – | – | Ref. [31] |
| Contact resistance | 25.1 | <2.6 | – | – | Ref. [32] |
| Contact resistance | 50.17 | <0.07 | – | – | Ref. [58] |
| Contact resistance | 8.5 | <12 | – | – | Ref. [29] |
| Contact resistance | 99.5 | 0.09–1 | <3 | Low | Ref. [60] |
| Resistance | 15.22 | <5 | <3 | – | Ref. [22] |
| Resistance | 0.0835 | 0.098–50 | – | – | Ref. [8] |
| Resistance | 0.048 | <18 | ~50 | – | Ref. [30] |
| Resistance | 0.011 | 1–120 | <3 | 24.8 | Ref. [16] |
| Resistance | 8.2 | <10 | <3 | Obvious | Ref. [11] |
| Resistance | ~2.75 | <4 | ~12 | Low | Ref. [52] |
| Resistance | ~2.07 | <15 | <3 | Obvious | Ref. [44] |
| Resistance | – | – | <3 | Low | Ref. [53] |
| Capacitive | ~0.02 | <22 | <3 | Low | Ref. [54] |
| Capacitive | 0.021 | <600 | – | – | Ref. [57] |
| Capacitive | ~0.1 | <10 | – | – | Ref. [15] |
| Capacitive | 44.5 | <0.1 | <3 | – | Ref. [59] |
| Capacitive | 0.00023 | <800 | ~100 | Obvious | Ref. [43] |
| Capacitive | ~13 | 0.5–5 | <3 | – | Ref. [65] |
| Capacitive | 0.55 | 0.2–7 | <3 | Obvious | Ref. [42] |
| Triboelectric | ~1.5 | 0.2–3.65 | – | Obvious | Ref. [55] |
| Triboelectric | ~0.45 | <20 | – | – | Ref. [56] |
| Triboelectric | ~44.14 | <0.75 | – | – | Ref. [62] |
| Triboelectric | ~0.046 | <170 | – | – | Ref. [64] |
| Triboelectric | ~0.011 | 0.07–40 | – | – | Ref. [38] |
| Piezoelectric | ~0.8 | 0.1–20.3 | – | – | Ref. [63] |

direct measurement, greatly reducing the back-end calibration processing. Moreover, the corresponding voltage of each pixel could be obtained by using the simulated displacement data and sensitivity curve of the sensor. The results show that the sensor arrays based on PET has large crosstalk, and even could not distinguish the stressed pixels, but the sensor arrays using the prslPDMS layer could accurately represent the stress point (Fig. 5c). The crosstalk isolation could be estimated by using $I_{so} = -20\log_{10}(\Delta V_{\text{stressedpixel}}/\Delta V_{\text{adjacentpixel}})$, and our sensor array is calculated as 32.14 dB, which is significantly better than the devices using PDMS (11.86 dB) or PET (5.38 dB) substrate. These results prove that the prslPDMS layer is suitable for avoiding crosstalk among adjacent pixels, which have obvious advantages to design the multilayer devices.

The sensor array was pressed with mold letter "B" to observe the pressure imaging performance, as shown in Fig. 5d. It could be found that the array without the spacer layer can hardly recognize patterns, but the letter "B" is clearly seen after using the prslPDMS layer and the crosstalk isolation is 33.41 dB. Hence, the voltage variation of pixels under pressure increases significantly, while the voltage of other pixels remain stable (Fig. 5e). Similar tests are carried out on the mold letter "I" and "N", which could also clearly exhibit the image of the embossed objects (More data could refer to the Supplementary Note 8). Table 1 illustrates the performance comparison with state-of-the-art pressure sensor, including sensitivity, detection range, stretchability, and crosstalk isolation[54–65], and the results reveal that the sensor array in this work achieve good properties in these fields (Supplementary Fig. 29). The application scenarios of the sensor array were

demonstrated in Fig. 5f, and its pixels were divided into four areas which represent different commands (including up, down, left and right). Attach the prepared device on hand, and the model movement in the game could be controlled by pressing the designated area, realizing a simple human–machine interaction (Supplementary movie 4). Further analysis was carried out on the crosstalk isolation performance of the prslPDMS layer in high spatial resolution devices. The above results show that high crosstalk isolation could be achieved when the ratio of spacer length to pixel length is 1:2, but an excessively long spacer is not conducive to fabricate the highly integrated devices. In practical applications, if higher density devices need to be designed and crosstalk isolation requirements are not strict, this ratio could be modified appropriately, such as 1:5, or even 1:10, etc. According to the mechanical simulation, in case of spacer length and pixel length being 1 μm and 5 μm, respectively, the crosstalk isolation is 22.43 dB and the pixel resolution could exceed 4000 ppi, which greatly satisfies the general detection accuracy. Therefore, the strain local confinement effect induced by micro-cage structure (prslPDMS) are beneficial for the preparation of ultrathin stretchable flexible electronics, which will have broad application prospects in precise tactile detection and high-resolution pressure imaging.

## Discussion

In summary, we demonstrated a high precision photo-reticulated strain localization films (prslPDMS) to fabricate the stretchable sensor arrays with ultralow spatial crosstalk. The micro-cage structure formed by prslPDMS layer acquires a strain local

confinement effect, which reduce the pixel deformation overflow by 90.3% compared with traditional flexible electronics. The prslPDMS was sandwiched between interdigital Ag NFs and micropatterned graphene layer, which not only ensures the stacked structure won't separate, but also provides the spacer for pressure sensing even in the bending state. The thickness of the sensor is only 60.59 µm with the help of PVA sacrificial layer, and its sensitivity or detection range could be adjusted by designing more rational structures. Hence, the sensor shows the sufficient pressure resolution to detect 1 g weight with a bending radius of about 5 mm, and it could monitor human pulse in different states (drinking, exercise, and rest) or analyze the grasping postures. The sensor arrays achieve the clear pressure imaging with the ultralow crosstalk of 33.41 dB, and it could be used to control the model movement. Furthermore, the simulation analysis illustrates that the devices still possess the high crosstalk isolation (22.43 dB) with the pixel resolution exceeding 4000 ppi, indicating that it has broad application in wearable electronics, soft robotics, and human–machine interaction.

## Methods

### Fabrication of Ag NFs interdigital electrodes

The PVA water solution (10 wt%) was poured into a 10 ml plastic syringe with a polished G10 needle. A syringe pump was used to control the flow rate of the solution (0.4 ml/h), and a constant potential (13 kV) was applied between the needle and circular grounded collectors to prepare the random PVA NFs. Subsequently, the PVA NFs was coated with silver via magnetron sputtering (PVD75 Kurt J. Lesker, Ar, 28 sccm, 100 W, 15 min) to acquire PVA/Ag NFs with concave or core/shell structure. Besides, the PVA water solution (5 wt%) was spin-coated on a glass substrate at 4000 r.p.m. for 60 s, and followed by baking at 100 °C for 120 s. PDMS (SYLGARD184 Dow Corning) was then spin-coated on the PVA/glass substrate at 6000 r.p.m. for 120 s and cured in the oven (120 °C, 20 min) to obtain the ultrathin PDMS layer. After the above procedures, PVA/Ag NFs were placed on the surface of the water, and use the ultrathin PDMS substrate to pick up the Ag NFs film. A positive photoresist (AZ5214) was spin-coated on Ag NFs, and the different masks will be fabricated via UV lithography. The uncovered Ag NFs will be etched by the 5 M dilute nitrate solution within 30 s, and the remaining photoresist could be washed away with acetone to acquire the desired Ag NFs electrodes.

### Fabrication of patterned prslPDMS layer

Benzophenone (Sigma-Aldrich) was dissolved in xylene in a weight ratio of 1:15 and added to the traditional PDMS mixture to form a homogeneous solution (3 wt%). Use a centrifuge to remove the air bubbles in the mixture at 40,000 r.p.m. for 10 min. The prslPDMS was then spin-coated on the desired substrate and exposed to UV radiation by using a portable UV lamp (14 mW/cm$^2$, 10 min). Benzophenone radicals will be generated, which will react with the silicon hydride groups in PDMS cross-linkers and the vinyl groups of PDMS monomers, thus preventing the traditional cross-linking reactions. A soft bake procedure was performed in a convection oven at 120 °C for approximately 150 s, and the unexposed PDMS will cure during the post exposure baking, while the exposed PDMS remains uncrosslinked and could be washed away in toluene. The thickness of the prslPDMS layer could be controlled by varying the spin speed or the dilution ratio, and the sample was rinsed in isopropanol and blown with N$_2$ gas.

### Fabrication the PDMS with pyramid microstructure

A silicon wafer was treated with plasma cleaner (O$_2$, 100 sccm, 100 W, 30 min) to enhance the adhesion between silicon substrate and photoresist. It was lithographically patterned by using S1813 positive photoresist, and the SiO$_2$ layer (700 nm) was sequentially deposited as a protective layer by using Plasma Enhanced Chemical Vapor Deposition (PECVD). Put the silicon wafer into etchant (33 wt% KOH and a little isopropanol), which was carried out in the thermostatic water bath to obtain the silicon template. Then, a small amount of trimethylchlorosilane (TMCS) was added to the surface of the silicon template for surface activation. Spin-coating PDMS mixture on the silicon template and remove air bubbles in a vacuum drying chamber. The PDMS with the pyramidal microstructure could be easily peeled-off after curing and another layer of self-assembled graphene is transferred on the microstructured PDMS. Finally, the microstructured PDMS with graphene is flipped over, and contacted with prslPDMS layer under gentle pressure for sensor assembly. Once the whole sensor was fabricated, it could be peeled off by dissolving the PVA sacrificial layer in water.

### Characterization and measurements

The morphologies of the samples were characterized by a field-emission scanning electron microscopy (Nova NanoSEM 450). In this work, a self-made measurement system was bulit to apply deformation and pressure, and the contact pressure was measured by a pressure sensor (Nano17 ATI). The DC power supply (Maynuo M8812) provides the voltage to sensor and a low-noise current preamplifier (MODEL SR570) was used to detect the current variation. A stepping motor (LinMot E1100) was carried out to measure the stability of the pressure sensor. Multichannel data acquisition systems (National Instrumental, PXIe-4300) were used to collect voltage signals to realize the image processing of the pressure distribution. A customized LabVIEW programs were used to process the data and control the movement of the objects.

### Reporting summary

Further information on research design is available in the Nature Portfolio Reporting Summary linked to this article.

## Data availability

All the data supporting the findings of this study are available within the main text and the Supplementary Information. The source data generated in this study are provided in the Source data file. Source data are provided with this paper.

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

## Acknowledgements

This work was supported by the National Key R & D Project from Ministry of Science and Technology, China (2021YFA1201603), the National Natural Science Foundation of China (Grant No. 51872031, 52073032, 61904013, 52125205, U20A20166, 61805015, and 61804011) and the Fundamental Research Funds for the Central Universities.

## Author contributions

J.Y.Z., Z.L.W., C.F.P., Y.F.Z., and Q.C.L. conceived the idea. J.Y.Z., C.F.P., Y.F.Z., and Q.C.L. designed the experiments. Y.F.Z., Q.C.L., J.H., Z.H.H., R.H.Z., X.H., and M.M.J. performed the experiments and analyzed the data. Y.F.Z., Q.C.L., C.F.P., J.Y.Z., and Z.L.W. wrote the paper. All authors discussed the results and commented on the manuscript.

## Competing interests

The authors declare no competing interests.
