## [Peer Review File · Nature Communications]

REVIEWER COMMENTS

Reviewer #1 (Remarks to the Author):

This manuscript deals with development of a tactile sensor with very high resolution. The work is interesting. It is more about making a device rather than addressing a science challenge. I did not see major breakthrough type science addressed here. I do not think Nature Communications is the right journal for this manuscript.

Reviewer #2 (Remarks to the Author):

This paper describes the development of a stretchable pressure sensor array with reduced spatial crosstalk by utilizing a microcage structure. In particular, the crosstalk was reduced by 90.3% by optimizing the sensor array structure. The paper also proposes several applications in consideration of social implementation, such as pulse detection. Achievement of both The multiple pixelization of stretchable sensor arrays and the suppression of crosstalk is one of the major issues for devices. The methods used to improve them and the actual realization of the devices were very good. However, there might be exist questions and points for improvement regarding the contents, and we describe how

1. Although the authors proposed the title "High-resolution Stretchable Pressure Sensor Array" in this paper, actually, little consideration was given to stretchability. If stretchability is to be demonstrated, it is necessary to examine the array when it is stretched by 30% to 50%. In addition, the durability of the arrays should be discussed when they are stretched 30% to 50% multiple times.
2. One of the advantages of this study is the use of PDMS as a substrate. Related to comment 1, one of the most serious problem of using a stretchable substrate such as PDMS for pressure sensors is the crosstalk between device strain and pressure rather than crosstalk between pixels (R. Matsuda et al. Scientific Reports 10(1) 2020). In particular, this study uses Ag-NFs. Is the device not affected by stretching when it is stretched by 30% to 50%?
3. There are many studies on stretchable pressure sensors (e.g. Y. Gao et al., Advanced Materials 29(39), 2017). Comparison regarding sensor sensitivity, comparison regarding strain (comment 2), comparison regarding pixels, etc., should be made in a multidimensional manner with other studies by using tables, graphs, etc.
4. It is assumed that Ag-NF means silver nanofibers. However, there is no explanation of its abbreviation.

5. The abstract states that " simulation analysis illustrates that the devices still attain high crosstalk isolation (22.43 dB) with the pixel resolution exceeding 4000 ppi". However, it should be excluded from the abstract because it is a simulation story and misleads readers in the case of this paper.
6. It is difficult to understand the analysis method for arraying. Are active or passive matrices used? Also, is current value (resistance value) used as detection? More detailed explanation is needed beyond Supplemental Fig. 14.
7. If the arraying is based on passive matrix and current value is used for detection, the exact current value for each pixel has not been detected without conversion (see R. Matsuda et al. Scientific Reports 10(1) 2020). This should be discussed in conjunction with comment
8. The text and figures in Fig. 1 are small and difficult to read. In particular, the structure and chemical formula in Fig. 1b are not visible. This should be improved.
9. Figures 3 g to i show a demonstration of a multi-point pressure sensor. What will it look like when the wiring is actually attached? Also, how is the wired connection made for the very thin sensors should be explained.
10. what is the resolution of prsIPDMS?
11. In general, there are no demonstration videos. Videos should be used as they are more persuasive.

Reviewer #3 (Remarks to the Author):

I have carefully read the manuscript from Zhang et al. reporting about the development of flexible pressure sensors arrays with limited cross talking effects.

The authors have developed a sort of microcage structure in order to avoid the mechanical cross talk between adjacent tactile cells, and most importantly have demonstrated that their pressure sensors are characterized by a very good sensitivity.

I found the developed approach interesting and I think that it can be potentially published in Nature Communications, but the authors should address the following points in order to have it accepted.

Comments:

1) The manuscript needs a more detailed benchmarking with already existing pressure/force sensor systems. I would suggest the authors to add a table reporting the recently developed systems and their performances to let the reader have a clear picture. Such analysis is reported in the SI, I think that in any case the performances of such sensors in comparison to already published works, must

be also highlighted in the main text. I would also clearly add in the main text a reference to the benchmarking plots reported in SI, just to let the reader understand where they can find this information.

2) How much the thickness of the grid walls impact in the performances of the sensor (sensing range) and mechanical crosstalk should be better highlighted. Are there any sort of design rules that have to be applied and followed to reproducibly fabricate such systems? Some hints are given in the main text and, again, more details are given in the SI, but I strongly suggest the authors to better highlight this point and give more details for the reader in the main text.

3) How much the reported procedure can be upscaled at low costs, for the routinely fabrication of such sensing systems over large areas?

4) The Ag electrodes configuration should be better explained, so far as I understood there is a common grounded electrodes and each pixel is measuring the current flowing between the second electrode and the common ground, is this correct?

5) Ag printed electrodes generally gets oxidized upon continuous exposure to air. The authors should address this point and demonstrated that the system behavior is stable and reproducible over time and has a sufficient life time for the envisaged applications. In fact, change of resistivity of the bottom electrodes will affect the overall pixel resistance, according to the scheme reported in th SI.

6) NO statistical analysis has been reported, please add details about it. How many devices and arrays have been measured? The authors must report all the graphs with the error bars, same for the reported sensitivities

7) I understand that bending of the substrate is not affecting the system sensitivity, which is a very important point. But I guess that transferring such system in a rough, uneven, substrate could dramatically change the sensitivity from pixel to pixel, how can the authors deal with this issue?

8) English must be strongly polished as there are several mistakes throughout the whole text.

Reviewer #4 (Remarks to the Author):

The authors propose a tactile sensor array with encapsulated patterning that is aimed at enhancing the spatial selectivity of the device. This design, in the aims of the authors, would reduce the cross-talk among adjacent taxels and thus improve the ability to localize tactile stimuli that are spatially distributed.

Overall, the proposed technology is sound and it deserves attention, however I would recommend addressing some issues as detailed hereafter.

First of all, the narrative grounded on the major need, in tactile sensing, to have transducers not affected by cross-talk is questionable. There are several reports, both in somatosensory neurophysiology and in biomimetic artificial touch studies, that point out that cross talk among adjacent sensors may be used as a tool to localize the stimulus thanks to triangulation mechanisms (and machine learning methods in some recent studies). Therefore, I would recommend revising the narrative of the paper and taking into account that cross-talk may also be beneficial to enhance the ability to localize the stimulus.

Considering the sectioning, I would suggest moving to the methods the subsections on device fabrication that are currently reported in the results section.

I would also recommend introducing a quantitative comparative discussion of results with respect to the metrological characteristics of pertinent state of the art sensors. Moreover, statistical analysis of experimental data should be added together with a presentation of the experimental protocols undertaken (including description of stimuli and strategy of stimuli administration, number of repetitions, and statistical indicators).

The supplementary materials are very good for the sake of reproducibility of the processes, however for the same purpose I would also suggest sharing experimental data and elaboration code in open manner by means of some kind of repository.

Minor aspects:

- The authors often use high resolution (e.g., high spatial resolution, high pressure resolution, ...), however in metrological terms better performance is associated with lower resolution; please revise somehow.
- In the section about "Structure and fabrication of ultralow crosstalk sensor" the authors suddenly introduce the horse and office logo patterning, however the reason why this is done is not clear. Please clarify.
- To quantitative analysis the elongation strain  to quantitative analyse the elongation strain
- Sensing sensitivity is redundant: maybe you can just use sensitivity
- The sentence "Furthermore, the sensor was carried out by the repetitive compression ..." is not clear

Point to Point Response to the referees' reports

(comments in black, responses in blue, changes highlighted in yellow):

Reviewer #1:

This manuscript deals with development of a tactile sensor with very high resolution. The work is interesting. It is more about making a device rather than addressing a science challenge. I did not see major breakthrough type science addressed here. I do not think Nature Communications is the right journal for this manuscript.

Response:

We would like to express our sincere thanks to the referee for her/his great effort to review the manuscript. As the reviewer suggested, we have reorganized the structure of our manuscript and strengthened the innovations, so as to facilitate readers to understand it. The main problem to be solved in this work is to effectively avoid inevitable mechanical crosstalk among adjacent pixels, which is important for multi-point detection in flexible electronics. When the device is subjected to intensive multi-point stimulation, if the deformation overflow cannot be well controlled, it will cause significant crosstalk, or even fail to distinguish exact stress point, which is not conducive to high-precision detection (**Fig. 5**). Considering this situation, we mimic the distribution of mechanoreceptors in human skin and introduce the *prs*/PDMS layer into the device, which could form the micro-cage structure and prevent the inevitable deformation diffusion. This enables the sensor array not only detect external stimuli independently, but also work together for a large area tactile perception. The scientific and technological innovations in this work are as follows:

1. A simple two-dimensional model is proposed to analyze the influence of different *prs*/PDMS designs on crosstalk isolation, including the four important parameters pixel length (l_p), spacer length (l_s) and thickness (t_s), and external displacement (D) (**Fig. 1c**). The simulation results show that the deformation overflow of device using *prs*/PDMS layer is reduced by 90.3% compared to that of conventional flexible electronics. The crosstalk isolation could exceed 25 dB with the ratio of spacer length to pixel length being 1: 2, which seems to us to achieve the better isolation effect. In practical applications, an excessively long spacer is not conducive to fabricate array devices. So if the higher density devices need to be prepared and crosstalk isolation requirements are not very strict, this ratio could also be appropriately increased, such as 1: 5, or even 1: 10, etc.

2. The *prs*/PDMS layer achieves the high transparency, stretchability and high precision patterning, which makes it possible to fabricate the multilayer devices. Furthermore, it could be used to encapsulate the devices with the interfacial toughness of about 55.94 J/m², so that the sensor acquires the sufficient pressure resolution even under bending or stretching conditions. Therefore, the sensor could detect the human pulse, analyze grasping postures and show the pressure distribution imaging. To sum up, we believe

that this is a feasible method for accurate tactile detection, which will have a profound impact on future tactile sensors.

The main text and the Supplementary Note are revised according to your comment, which is listed as following for your convenience (all changes made in the revised MS is highlighted in yellow):

Main text, Figure 1, Page 29

Fig. 1 | Principle of ultralow crosstalk sensor with micro-cage structure based on prs/PDMS layer. **a** Schematic illustration of various independent mechanoreceptors in glabrous skin for accurate tactile perception. **b** Proposed chemistry of benzophenone inhibiting PDMS crosslinking under UV light to prepare prs/PDMS layer. **c** Simple geometric analysis of micro-cage structure formed by prs/PDMS under external pressure. **d** Two-dimensional deformation simulation analysis of the PDMS & prs/PDMS and PET under external pressure, and the micro-cage structure of the former enables the small elongation strain compared with the latter. **e** Quantitative analysis of displacement variation along the x-axis for different models, and the displacement of adjacent pixel using prs/PDMS layer decrease by 90.3% compared with PET. **f** Crosstalk isolation versus the ratio of spacer length to pixel length, the better isolation effect could be achieved when its value is greater than 1 : 2.

Main text, Page 6-7

Design concept of micro-cage structure based on *prs*/PDMS layer. Human sense of touch deals with spatiotemporal perception under external stimuli through a large number of receptors (Fig. 1a). The relevant information reaches the spinal cord through multiple nerves and is transmitted to the central nervous system via two main pathways for higher-level processing and interpretation: spinothalamic and dorsal-column-medial-lemniscal. The latter could quickly convey pressure/vibration information to the brain for precise tactile detection. Taking fingertips as an example, there are many mechanoreceptors embedded in skin at different depths, which are mainly divided into four categories: slow-adapting receptors responding to static pressures (SA-I and SA-II) and fast-adapting receptors responding to dynamic forces or vibration (FA-I and FA-II). Some of these mechanoreceptors are distributed independently at the intermediate ridges between the epidermis and dermis, which could produce stress concentration to enhance pressure perception. Our pressure sensors mimic this structure and individual receptors, enabling each sensor to operate with ultralow crosstalk. As shown in Fig. 1b, doping benzophenone into PDMS will inhibit its crosslinking when exposed to UV light, thus forming the photo-reticulated PDMS (Supplementary Fig. 1). The micro-cage structure is formed after encapsulating with another layer of electrodes, and pressure sensor could be prepared within the cage. Besides, the boundary of the micro-cage composed by photo-reticulated PDMS could separate different sensors and prevent inevitable deformation diffusion, showing the effect of strain local confinement, so it is called photo-reticulated strain localization PDMS (*prs*/PDMS). A simple two-dimensional model is proposed to analyze the deformation of micro-cage structure under external pressure (Fig. 1c). It could be found that the entire model has four key parameters, which are pixel length (l_p), spacer length (l_s) and thickness (t_s), and external displacement (D). When a pixel is subjected to external pressure, both the top electrode and *prs*/PDMS spacer will deform, thereby gradually expanding to adjacent pixel, resulting in mechanical crosstalk. Deformation simulation analysis is performed on PDMS model with *prs*/PDMS spacer and the model only with PET (Fig. 1d). The results demonstrate that the former realizes the small elongation strain due to the strain confinement of *prs*/PDMS spacer, but PET substrate with high toughness and non-stretchable characteristics shows the large deformation. Moreover, a coordinate system is established on the right boundary of stressed pixel to quantitative analyze the elongation strain along the x-axis and y-axis (Fig. 1e). The displacement along the y-axis declines slowly for the PET model, and its average displacement in adjacent pixel is 6.13 μm . However, the deformation of *prs*/PDMS spacer model decreases rapidly in the spacer layer region, and the average displacement of adjacent pixel is only 0.56 μm , which decreases by 90.3% compared with PET substrate (the maximum deformation of stressed pixel). Figure 1f describes the crosstalk isolation versus the different ratios of spacer length to pixel length, and its value reaches 25.03 dB with the ratio of 0.5, which can be considered as better mechanical crosstalk isolation (More detailed analysis could refer to the Supplementary Fig. 10).

Main text, Page 7-8

Structure of pressure sensor array. Figure 2a shows the exploded view of the stretchable ultralow crosstalk sensor array. The device mainly consists of three parts:

the patterned Ag NFs interdigital electrodes, the patterned *prsi*/PDMS layer and graphene attached to the PDMS with pyramid microstructures. The main challenge in this work is to prepare the patterned dielectric films (*prsi*/PDMS), which can not only separate adjacent pixels to form micro-cage structure, but also show the function of adhesion and support for packaging devices. SEM images of *prsi*/PDMS layer with different resolutions are shown in Fig. 2b, with a precision up to 100 μm . Then the patterning effect on different substrates was also verified, such as glass (horse) and silicon (office logo), indicating its excellent adaptability. In addition, the stretchability and transmittance of *prsi*/PDMS layer were further demonstrated in Fig. 2c. The results show the *prsi*/PDMS layer possesses the similar stretchability ($\sim 5.08\text{MPa}$) and transmittance (91.85% in visible light) with PDMS, allowing the preparation of transparent stretchable devices. For more detailed fabrication process, please refer to the Method, Supplementary Note 1 – 3 and Supplementary Movie 1. Since the high precision transparent stretchable *prsi*/PDMS film could achieve strain local confinement, it can be used to prepare multilayer devices and improve the mechanical stability, which could provide a solid foundation for more sophisticated electronics (Fig. 2d). Figure 2e and 2f show the 6×6 sensor array is well attached on the palm no matter in the flat or curly state. The insert figure (Fig. 2e) exhibits the single sensor on fingertip ($2.0 \times 2.0 \text{ mm}^2$) with a transmittance of 50.36%, and the enlarged cross-section SEM image (Fig. 2f) shows the thickness of multilayer stacked structure is only 60.59 μm .

Main text, Page 30

Fig. 2 | Structure and design concept of the pressure sensor. a Schematic illustration of the structure of pressure sensor arrays. **b** SEM images of *prsi*/PDMS layer with different resolutions, which could reach 100 μm ; The right photographs demonstrate the patterns on various substrates (top: glass, bottom: silicon wafer), indicating its good

adaptability. **c** Tensile properties (left) and UV-visible spectra (right) of *prs*/PDMS exhibit the excellent stretchability and transparency. **d** Schematic showing the design concept for the ultralow crosstalk sensor, where the *prs*/PDMS layer perform the effect of strain local confinement for mechanical stability as well as multilayer device. Optical photographs of sensor arrays attached on palm no matter in the flat (**e**) or curly (**f**) state, which achieve the transparency of 50.36 % and thickness of 60.59 μm .

Reviewer #2:

This paper describes the development of a stretchable pressure sensor array with reduced spatial crosstalk by utilizing a microcage structure. In particular, the crosstalk was reduced by 90.3% by optimizing the sensor array structure. The paper also proposes several applications in consideration of social implementation, such as pulse detection. Achievement of both the multiple pixelization of stretchable sensor arrays and the suppression of crosstalk is one of the major issues for devices. The methods used to improve them and the actual realization of the devices were very good. However, there might be exist questions and points for improvement regarding the contents, and we describe how

Answers:

We would like to express our sincere thanks to the referee for her/his great effort to review the manuscript and positive evaluation on our work.

1. Although the authors proposed the title "High-resolution Stretchable Pressure Sensor Array" in this paper, actually, little consideration was given to stretchability. If stretchability is to be demonstrated, it is necessary to examine the array when it is stretched by 30% to 50%. In addition, the durability of the arrays should be discussed when they are stretched 30% to 50% multiple times.

Response:

Thanks the reviewer for the suggestion. It is essential to demonstrate the performance of sensor arrays under stretching, which could better meet the practical applications of wearable electronics. As the reviewer suggested, the arrays were stretched by 50% and subjected to cyclic stretching about 100 times. The results show that the voltage of the stressed pixels in the array decrease after being stretched, which may be attributed to strain that destroys the integrity of bottom interdigital electrodes and graphene electrodes, but it could still reflect the pressure distribution of the embossed objects. It is worth noting that its properties remain similar to the initial state even after 100 times of cyclic stretching, indicating the good durability.

The main text and the Supplementary Note are revised according to your comment, which is listed as following for your convenience (all changes made in the revised MS is highlighted in yellow):

Supplementary Note 8, Supplementary Figure 24, Page S35

When the device is subjected to 50% tensile strain, the voltage measured at each pixel drops significantly, which means that the stretching will increase the resistance of each pixel. Although the tactile sensing performance has decreased, the pattern of the letter "B" still could be observed. Furthermore, when the deformation is released after

100 times of cyclic stretching, its performance roughly returns to the original state, indicating the good durability. The current variation shown below is also consistent with the above results, indicating that the sensor array realizes an excellent stability.

Supplementary Figure 24 | Imaging performance of sensor array under tensile strain.

2. One of the advantages of this study is the use of PDMS as a substrate. Related to comment 1, one of the most serious problems of using a stretchable substrate such as PDMS for pressure sensors is the crosstalk between device strain and pressure rather than crosstalk between pixels (R. Matsuda et al. Scientific Reports 10(1) 2020). In particular, this study uses Ag-NFs. Is the device not affected by stretching when it is stretched by 30% to 50%?

Response:

Thanks the reviewer for the suggestion. It is important to distinguish the crosstalk caused by strain and external pressure in order to achieve the accurate tactile detection. And it just reflects the importance of the micro-cage structure formed by *pr*s/PDMS layer in our sensor, which shows the function of adhesion and support to ensure that the strain won't significantly affect the pressure sensing properties. Furthermore, we further investigated the effect of strain on single sensor and sensor array, and the sensitivity curve of single sensor gradually decreases with the larger strain. Taking the linear region as an example, the sensitivity changes from 2.23 kPa^{-1} to 0.68 kPa^{-1} when the strain increases to 50%. After the deformation is removed, its properties could also be restored to the initial state (Fig. 3f). As shown in Question 1, although the performance of sensor array declines under 50% strain, it could still show the pressure distribution. The pixels receiving pressure exhibit the obvious voltage variation, while others remain basically unchanged.

The main text and the Supplementary Note are revised according to your comment, which is listed as following for your convenience (all changes made in the revised MS is highlighted in yellow):

Main text, Page 10

Further tests were carried out to demonstrate the stretchability of the pressure sensor, and it could be found that the sensitivity gradually decreases from 2.23 kPa^{-1} to 0.68 kPa^{-1} with the deformation increasing to 50% (linear region). However, its performance return to the initial state after releasing the deformation, suggesting the excellent resilience (Fig. 3f).

Fig. 3 | Pressure sensing performance of the ultrathin sensor. **a** Schematic diagram of the experimental setup for pressure sensing. **b** Sensitivity curve of the sensor (top); Simulation of stress distribution of the *pr*s/PDMS layer and microstructured PDMS under external pressure (middle); Cross-sectional SEM images of the sensor under different pressures (bottom). Relative change in current ($\Delta I/I_0$) of the sensors with different interdigital Ag NFs electrodes (c), spacer layer thickness (d) and pyramid sizes (e). Relative change in current ($\Delta I/I_0$) of the sensor with different strains (f) and bending radii (g). Inset: sensor response after recovery from deformations (left) and to tiny forces even in the bending state (right). **h** Stability measurement of the pressure sensor, and the cycle period is over 5000 cycles. Inset: The response performance of the sensor with different spacer thickness.

Supplementary Note 5, Supplementary Figure 15, Page S24

Supplementary Figure 15 | Sensitivity (a) and detection limit (b) of sensors under different deformations.

It could be seen that the sensitivity of the sensor gradually decreases with the larger strain. Taking the linear region as an example, the sensitivity changes from 2.23 kPa⁻¹ to 0.68 kPa⁻¹ when the strain increases to 50%. Additionally, the lower detection limit has also increased significantly, from ~500 Pa to ~90 kPa. Tensile strain will destroy the integrity of the interdigital Ag NFs or Gr electrode, so more loops with good conduction are formed only after more pressure is applied, leading to the decreased sensitivity and lower detection limits. However, its tactile detection performance could return to the original state when the strain is released.

3. There are many studies on stretchable pressure sensors (e.g. Y. Gao et al., *Advanced Materials* 29(39), 2017). Comparison regarding sensor sensitivity, comparison regarding strain (comment 2), comparison regarding pixels, etc., should be made in a multidimensional manner with other studies by using tables, graphs, etc.

Response:

Thanks the reviewer for the suggestion. As the reviewer suggested, the performance comparison with other existing sensors is illustrated in **Table 1**. Furthermore, two representative works are selected to reflect the advantages of our sensor, including stretchability, thickness, sensitivity, transparency, detection range and crosstalk isolation (**Supplementary Fig. 29**). It could be seen that the sensor array achieves good properties in these fields, indicating the broad application perspective in wearable electronics.

The main text and the Supplementary Note are revised according to your comment, which is listed as following for your convenience (all changes made in the revised MS is highlighted in yellow):

Main text, Table 1, Page 34

Table 1 Performance comparison with state-of-the-art

pressure sensor

Sensing Mechanisms	Sensitivity/kPa ⁻¹	Detection range/kPa	Stretchability/%	Crosstalk isolation/dB	Studies
Contact resistance	18.94 (10 μm) 4.04 (30 μm)	< 40 < 200	~ 50	~ 33.41	This work
Contact resistance	4.88	0.37 ~ 8	~ 40	--	Ref. 61
Contact resistance	0.7	< 25	< 3	Low	Ref. 28
Contact resistance	~ 15.1	0.2 ~ 59	--	--	Ref. 31
Contact resistance	25.1	< 2.6	--	--	Ref. 32
Contact resistance	50.17	< 0.07	--	--	Ref. 58
Contact resistance	8.5	< 12	--	--	Ref. 29
Contact resistance	99.5	0.09 ~ 1	< 3	Low	Ref. 60
Resistance	15.22	< 5	< 3	--	Ref. 22
Resistance	0.0835	0.098 ~ 50	< 3	--	Ref. 8
Resistance	0.048	< 18	~ 50	--	Ref. 30
Resistance	0.011	1 ~ 120	< 3	24.8	Ref. 15
Resistance	8.2	< 10	< 3	Obvious	Ref. 9
Resistance	~ 2.75	< 4	~ 12	Low	Ref. 52
Resistance	~ 2.07	< 15	< 3	Obvious	Ref. 44
Resistance	--	--	< 3	Low	Ref. 53
Capacitive	~ 0.02	< 22	< 3	Low	Ref. 54
Capacitive	0.021	< 600	< 3	--	Ref. 57
Capacitive	~ 0.1	< 10	< 3	--	Ref. 14
Capacitive	44.5	< 0.1	< 3	--	Ref. 59
Capacitive	0.00023	< 800	~ 100	Obvious	Ref. 43
Capacitive	~ 13	0.5 ~ 5	< 3	--	Ref. 65
Capacitive	0.55	0.2 ~ 7	< 3	Obvious	Ref. 42
Triboelectric	~ 1.5	0.2 ~ 3.65	< 3	Obvious	Ref. 55
Triboelectric	~ 0.45	< 20	--	--	Ref. 56
Triboelectric	44.14	< 0.75	--	--	Ref. 62
Triboelectric	~ 0.046	< 170	--	--	Ref. 64
Triboelectric	~ 0.011	0.07 ~ 40	--	--	Ref. 38
Piezoelectric	~ 0.8	0.1 ~ 20.3	--	--	Ref. 63

Supplementary Note 8, Supplementary Figure 29, Page S39

Supplementary Figure 29 | Performance comparison with state-of-the-art pressure sensors.

Comparison of our sensor with existing sensors in terms of sensitivity and detection range are depicted in Supplementary Fig. 29¹⁻²⁶. Although it is not the most sensitive, it shows a wider linear response range. The two representative works are selected to further analyze the advantages of our sensor (shown on right), including stretchability, thickness, sensitivity, transparency, detection range and crosstalk isolation. It could be seen that our sensor has good performance in all aspects, which can be attached to human skin with large scale for precise tensile detection.

4. It is assumed that Ag-NF means silver nanofibers. However, there is no explanation of its abbreviation.

Response:

Thanks the reviewer for the suggestion. As the reviewer suggested, we modified the ambiguous words in the MS with a rigorous expression.

The main text and the Supplementary Note are revised according to your comment, which is listed as following for your convenience (all changes made in the revised MS is highlighted in yellow):

Main text, Page 9

Figure 3c illustrates the relative change in current ($\Delta I/I_0$) versus applied pressure for a series of sensor with different interdigital silver nanofibers (Ag NFs) electrodes.

5. The abstract states that " simulation analysis illustrates that the devices still attain high crosstalk isolation (22.43 dB) with the pixel resolution exceeding 4000 ppi". However, it should be excluded from the abstract because it is a simulation story and misleads readers in the case of this paper.

Response:

Thanks the reviewer for the suggestion. As the reviewer suggested, we have modified the abstract with more rigorous expression.

The main text and the Supplementary Note are revised according to your comment, which is listed as following for your convenience (all changes made in the revised MS is highlighted in yellow):

Main text, Page 2

Tactile sensors with high spatial resolution and low crosstalk are crucial to manufacture large scale flexible electronics. However, denser arrays mean greater mechanical crosstalk among adjacent pixels, which significantly impair the detection accuracy. Here, we demonstrated the photo-reticulated strain localization films (*prs*/PDMS) to prepare the ultralow crosstalk sensor array, which form a micro-cage structure to reduce the pixel deformation overflow by 90.3% compared to that of conventional flexible electronics. It is worth noting that *prs*/PDMS acts as an adhesion layer and provide spacer for pressure sensing. Hence, the sensor achieves the sufficient pressure resolution to detect 1g weight even in bending condition, and it could monitor human pulse under different states or analyze the grasping postures. Experiments show that the sensor array acquires clear pressure imaging with the ultralow crosstalk (33.41 dB), indicating that it has a broad application prospect in precise tactile detection.

6. It is difficult to understand the analysis method for arraying. Are active or passive

matrices used? Also, is current value (resistance value) used as detection? More detailed explanation is needed beyond Supplemental Fig. 14.

Response:

Thanks the reviewer for the suggestion. The passive matrices are used in this work. Since the multi-channel data acquisition card (PXIe-4300, National Instruments) in our lab could only detect voltage, we connect an external resistor in series with each sensor to form a loop with power supply. By measuring the voltage of the resistance, the current in the whole circuit could be deduced. According to the Kirchhoff's law, the current in the loop is equal everywhere, so this current could be considered as the current flowing through the sensor. When the pressure on the sensor increases, its resistance will gradually decline. And then more current will be generated in the loop, which will increase the voltage of the external resistor, so we choose this voltage to reflect the real-time pressure distribution of the sensor array. Multiple sensors for grasping posture analysis also work on this principle, except that multiple sensors need to lead out two electrodes respectively (**Supplementary Fig. 18**), whereas the structure of array has a common port (**Supplementary Fig. 22**).

The main text and the Supplementary Note are revised according to your comment, which is listed as following for your convenience (all changes made in the revised MS is highlighted in yellow):

Main text, Page 12

Multiple sensors are attached on the finger to analyze the grasping posture, as shown in Fig. 4g. 14 sensors are stucked on the phalanges of the palm, and the fine copper wires with a diameter about 30 μm was used to lead out the electrodes (Supplementary Fig. 18). In order to facilitate multi-channel data acquisition, each sensor is connected in series with an external resistor. By detecting the voltage of resistors, the current variation in the circuit could be calculated, thereby reflecting the pressure on the sensor. More detailed analysis could refer to the Supplementary Fig. 19. Figure 4h reveals the data when an apple is half held, the sensors at the fingertips show the larger voltage, but all sensors have the high voltage with the apple fully grasped (Fig. 4i, Supplementary Fig. 20 and Supplementary Moive 3). These results are consistent with the actual situation, so our ultrathin sensors possess the excellent performance in both single sensor precise measurement and collaborative detection of multiple devices.

Supplementary Note 7, Supplementary Figure 18, Page S28

Supplementary Figure 18 | Measurement principle with multiple sensors attached to the palm.

Multiple sensors attached to the palm is described in Supplementary Fig. 18a. Each sensor is fixed with scotch tape, then connected in series to the loop with alligator clips. The sensor in this work is very thin, so the fine copper wires ($\sim 30 \mu\text{m}$) are used to lead out the electrodes, and silver paste is used to connect the interdigital electrodes and wires. Additionally, use PDMS to encapsulate the silver paste to prevent it from peeling off (Supplementary Fig. 18b). Supplementary Figure 18c demonstrates the principle of simultaneous measurement of multiple sensors. Since the data acquisition card (PXIe-4300, National Instruments) in the lab could only collect the voltage, we connect an external resistor in series with each sensor. By measuring the voltage of the resistance, the current flowing through it could be calculated. According to the Kirchhoff's law, the current in the entire loop is equal everywhere, so this current could be considered as the current flowing through the sensor. Therefore, 14 sensors need 14 external resistors, and 14 loops are connected in parallel with each other.

Supplementary Note 8, Supplementary Figure 22, Page S33

Supplementary Figure 22a demonstrates the schematic diagram of a 6×6 sensor array for pressure detection. The electrical signal of the pixels will change with the objects placed on its surface, thereby real-time presenting the shape of the object. The measurement principle here is similar to that of the previous multiple sensors, and each pixel in the array is connected to a resistor. Multi-channel synchronous data acquisition card collects the voltage on the resistor, then calculates the pressure on the pixel, which is equivalent to multi-electrometer scanning simultaneously.

Supplementary Figure 22 | Multi-channel data measurement system. (a) Schematic illustration of the pressure mapping process (left). Optical image of the data measurement system (right). (b) The principle of the multi-channel synchronous data acquisition.

7. If the arraying is based on passive matrix and current value is used for detection, the exact current value for each pixel has not been detected without conversion (see R. Matsuda et al. Scientific Reports 10(1) 2020). This should be discussed in conjunction with comment.

Response:

Thanks the reviewer for the suggestion. According to the above analysis, the current of each sensor could be calculated by the ratio of the voltage of the external resistors to its resistance. As the reviewer suggested, we further converted the measured voltage into the exact current (**Supplementary Fig. 20 and Supplementary Fig. 24**). In addition, the voltage measured varies with different resistors. For the smaller resistance, although the current variation in the loop is relatively large, its voltage will be low, and the measured pressure range will also be reduced. The larger external resistor means the wider voltage detection range, and the data is easy to observe. Therefore, the external resistor in this work is chosen to be 25 k Ω (**Supplementary Fig. 19**).

The main text and the Supplementary Note are revised according to your comment, which is listed as following for your convenience (all changes made in the revised MS is highlighted in yellow):

Supplementary Note 7, Supplementary Figure 19-20, Page S29-31

Supplementary Figure 19a depicts the equivalent circuit of single-channel signal acquisition. Based on the Kirchhoff's law, the current of the sensor could be obtained by the following formula:

$$I_s = I_r = \frac{U_r}{R_r} \quad (7-1)$$

I_s : the current of pressure sensor, I_r : the current of external resistance, U_r : the voltage of external resistance; Furthermore, the voltage of external resistance could also be deduced as follows:

$$U_r = \frac{R_r}{R_s + R_r} U_s \quad (7-2)$$

U_s : the voltage of power supply;

According to previous measurement data, the empirical formula of the current on sensor with the pressure could be fitted (Supplementary Fig. 19b), then the relationship between its resistance and pressure could also be calculated (Supplementary Fig. 19c):

$$I_s = 3.9 \times 10^{-9} P - 4.0 \times 10^{-5} \quad (7-3)$$

$$R_s = 89921.3 \exp\left(\frac{-P}{19196.0}\right) + 1913.4 \quad (7-4)$$

Subsequently, the influence of different external resistors on loop current and measurement voltage U_r is further studied. It could be found that the loop current gradually declines with the larger external resistance, but the range of U_r increases slightly which is beneficial for signal acquisition.

Supplementary Figure 19 | The conversion principle of current and voltage in multi-channel data acquisition. (a) Equivalent circuit diagram of single-channel signal acquisition. Fitted curves of current (b) and resistance (c) of the sensor in the linear region of 20 to 200 kPa. Current variation (d) and the voltage variation on external resistance (e) with different resistors in series to the loop.

The power supply used in this experiment is 10 V, and the external resistance is 25 kΩ. According to the above theoretical analysis, the pressure and current of the sensor could be calculated by the following formulas:

$$I_s = \frac{U_r}{25000} \quad (7-5)$$

$$U_r = \frac{250000}{89921.3 \exp\left(\frac{-P}{1919.6}\right) + 26913.4} \quad (7-6)$$

It could be found that the current and pressure both show the same variation trend, which is consistent with actual situation, so we choose the voltage (U_r) to reflect the real-time pressure distribution.

Supplementary Figure 20 | The current and pressure signal diagram of multiple sensors in different grip postures.

Supplementary Note 8, Supplementary Figure 24, Page S35

Supplementary Figure 24 | Imaging performance of sensor array under tensile strain.

When the device is subjected to 50% tensile strain, the voltage measured at each pixel drops significantly, which means that the stretching will increase the resistance of each pixel. Although the tactile sensing performance has decreased, the pattern of the letter “B” could still be observed. Furthermore, when the deformation is released after 100 times of cyclic stretching, its performance roughly returns to the original state, indicating the good durability. The current variation shown below is also consistent with the above results, indicating that the sensor array realizes an excellent stability.

8. The text and figures in Fig. 1 are small and difficult to read. In particular, the structure and chemical formula in Fig. 1b are not visible. This should be improved.

Response:

Thanks the reviewer for the suggestion. As the reviewer suggested, we put the structure and chemical formula in the Supplementary Information (**Supplementary Fig. 1**), only leaving the key part in the main text (**Fig. 1b**). Benzophenone will produce free radicals under UV irradiation, which will inhibit PDMS crosslinking and form *pr*s/PDMS layer.

The main text and the Supplementary Note are revised according to your comment, which is listed as following for your convenience (all changes made in the revised MS is highlighted in yellow):

Main text, Page 34

Fig. 1 | Principle of ultralow crosstalk sensor with micro-cage structure based on *prs*/PDMS layer. **a** Schematic illustration of various independent mechanoreceptors in glabrous skin for accurate tactile perception. **b** Proposed chemistry of benzophenone inhibiting PDMS crosslinking under UV light to prepare *prs*/PDMS layer. **c** Simple geometric analysis of micro-cage structure formed by *prs*/PDMS under external pressure. **d** Two-dimensional deformation simulation analysis of the PDMS & *prs*/PDMS and PET under external pressure, and the micro-cage structure of the former enables the small elongation strain compared with the latter. **e** Quantitative analysis of displacement variation along the x-axis for different models, and the displacement of adjacent pixel using *prs*/PDMS layer decrease by 90.3% compared with PET. **f** Crosstalk isolation versus the ratio of spacer length to pixel length, the better isolation effect could be achieved when its value is greater than 1:2.

Supplementary Note 1, Supplementary Figure 1, Page S1-2

Supplementary Figure 1 | Preparation process and proposed chemistry of photo-reticulated strain localization PDMS.

The photo-reticulated strain localization PDMS (*prs*/PDMS) was selected as the spacer layer due to its low cost, operation in ambient light and simple fabrication process. The conventional PDMS (SYLGARD184 Dow Corning) consists of the repeating $-\text{OSi}(\text{CH}_3)_2-$ units. The PDMS base monomer is vinyl terminated, while the crosslinking monomers are methyl terminated which contain silicon hydride $-\text{OSiHCH}_3-$ units. The PDMS monomers crosslink via a reaction between the monomer vinyl groups and the crosslinker silicon hydride groups to form $\text{Si}-\text{CH}_2-\text{CH}_2-\text{Si}$ linkages during curing. However, the benzophenone radical is formed when it is irradiated of $\text{UV} < 365 \text{ nm}$. And these radicals react with the silicon hydride groups presented in the PDMS crosslinkers and the vinyl groups of the PDMS monomers, which will prevent PDMS from undergoing the traditional crosslinking reactions. During the post exposure bake, the unexposed PDMS gets cured, while the exposed PDMS could be washed away in toluene. The detailed fabrication process is described below:

1. The PDMS was mixed thoroughly at a weight ratio of 1: 10 and stirred it to obtain a homogeneous solution. Then, the benzophenone was dissolved in xylene at a weight ratio of 1: 10 and added to the conventional PDMS mixture to yield the *prs*/PDMS solution with desired concentration.

2. The *prs*/PDMS mixture was spin-coated on the desired substrate, and the thickness of the *prs*/PDMS layer could be controlled by varying the spin speed.

3. The *prs*/PDMS layer was selectively exposed to $\text{UV} < 365 \text{ nm}$ with the exposure power of $14 \text{ mW}/\text{cm}^2$ through a chrome mask for 10 min. Either a traditional lithography machine or a portable UV lamp could be used for exposure. A proximity exposure at a distance of $\sim 100 \text{ um}$ was used.

4. The samples should be put in a convection oven at $120 \text{ }^\circ\text{C}$ for approximately 150 s after the exposure processing. Put the sample in the toluene for 3 ~ 5 s to wash off the exposed regions and it will be rinsed with isopropanol and blown with N_2 gas.

After the encapsulation of *prs*/PDMS layer, the microstructured PDMS with graphene was laminated on its surface for sensor assembly.

9. Figures 3 g to i show a demonstration of a multi-point pressure sensor. What will it look like when the wiring is actually attached? Also, how is the wired connection made for the very thin sensors should be explained.

Response:

Thanks the reviewer for the suggestion. As the reviewer suggested, we further explain how to lead to the electrodes in the **Supplementary Fig. 18**. The fine copper wires used in this work are about 30 μm in diameter, and the silver paste is used to connect interdigital electrodes and copper wires. After drying the silver paste, use PDMS to encapsulate the joints to prevent it from breaking.

The main text and the Supplementary Note are revised according to your comment, which is listed as following for your convenience (all changes made in the revised MS is highlighted in **yellow**):

Supplementary Note 7, Supplementary Figure 18, Page S28

Supplementary Figure 18 | Measurement principle with multiple sensors attached to the palm.

10. what is the resolution of *prs*/PDMS?

Response:

Thanks the reviewer for the suggestion. The resolution of *prs*/PDMS in this work could

reach about 100 μm , and the SEM images are shown in Fig. 2b and Supplementary Fig. 5. Furthermore, the patterning effect could be realized on different substrates, indicating its excellent adaptability.

The main text and the Supplementary Note are revised according to your comment, which is listed as following for your convenience (all changes made in the revised MS is highlighted in yellow):

Main text, Page 30

Fig. 2 | Structure and design concept of the pressure sensor. **a** Schematic illustration of the structure of pressure sensor arrays. **b** SEM images of *prs*/PDMS layer with different resolutions, which could reach 100 μm ; The right photographs demonstrate the patterns on various substrates (top: glass, bottom: silicon wafer), indicating its good adaptability. **c** Tensile properties (left) and UV-visible spectra (right) of *prs*/PDMS exhibit the excellent stretchability and transparency. **d** Schematic showing the design concept for the ultralow crosstalk sensor, where the *prs*/PDMS layer perform the effect of strain local confinement for mechanical stability as well as multilayer device. Optical photographs of sensor arrays attached on palm no matter in the flat (**e**) or curly (**f**) state, which achieve the transparency of 50.36 % and thickness of 60.59 μm .

Main text, Page 8

SEM images of *prs*/PDMS layer with different resolutions are shown in Fig. 2b, with a precision up to 100 μm . Then the patterning effect on different substrates was also verified, such as glass (horse) and silicon (office logo), indicating its excellent adaptability.

Supplementary Note 1, Supplementary Figure 5, Page S7

Supplementary Figure 5 shows the patterned *prs*/PDMS film with different precisions ranging from 1 mm to 100 μm . The *prs*/PDMS film gradually changes from square holes to round holes with the increasing exposure resolution. In addition, it was found that the accuracy of 50 μm could be achieved, but its uniformity is poor, so we believe that the resolution of *prs*/PDMS film could reach 100 μm .

Supplementary Figure 5 | The SEM images of *prs*/PDMS with different resolutions.

11. In general, there are no demonstration videos. Videos should be used as they are more persuasive.

Response:

Thanks the reviewer for the suggestion. As the reviewer suggested, we have added four videos. 1. Electrospinning and graphene preparation; 2. Pulse detection under different pressures; 3. Multiple sensors used for grasping posture analysis; 4. Demo of application scenarios.

The main text and the Supplementary Note are revised according to your comment, which is listed as following for your convenience (all changes made in the revised MS is highlighted in yellow):

Main text, Page 8

The results show the *prs*/PDMS layer possesses the similar stretchability ($\sim 5.08\text{MPa}$) and transmittance (91.85% in visible light) with PDMS, allowing the preparation of transparent stretchable devices. For more detailed fabrication process, please refer to the Method, Supplementary Note 1 – 3 and Supplementary Movie 1.

Main text, Page 11

The sensor exhibits the larger base current with the increasing pressure, and it also reveals a stable current fluctuation which contains the pulse information of the human body (Supplementary moive 2).

Main text, Page 13

Figure 4h reveals the data when an apple is half held, the sensors at the fingertips show the larger voltage, but all sensors have the high voltage with the apple fully grasped (Fig. 4i, Supplementary Fig. 20 and Supplementary Movie 3).

Main text, Page 15

Attach the prepared device on hand, and the model movement in the game could be controlled by pressing the designated area, realizing a simple human-machine interaction (Supplementary movie 4).

Reviewer #3:

I have carefully read the manuscript from Zhang et al. reporting about the development of flexible pressure sensors arrays with limited cross talking effects.

The authors have developed a sort of microcage structure in order to avoid the mechanical cross talk between adjacent tactile cells, and most importantly have demonstrated that their pressure sensors are characterized by a very good sensitivity.

I found the developed approach interesting and I think that it can be potentially published in Nature Communications, but the authors should address the following points in order to have it accepted.

Answers:

We would like to express our sincere thanks to the referee for her/his great effort to review the manuscript and positive evaluation on our work.

1. The manuscript needs a more detailed benchmarking with already existing pressure/force sensor systems. I would suggest the authors to add a table reporting the recently developed systems and their performances to let the reader have a clear picture. Such analysis is reported in the SI, I think that in any case the performances of such sensors in comparison to already published works, must be also highlighted in the main text. I would also clearly add in the main text a reference to the benchmarking plots reported in SI, just to let the reader understand where they can find this information.

Response:

Thanks the reviewer for the suggestion. As the reviewer suggested, the performance comparison with other existing sensors is illustrated in **Table 1**. Furthermore, two representative works are selected to reflect the advantages of our sensor, including stretchability, thickness, sensitivity, transparency, detection range and crosstalk isolation (**Supplementary Fig. 29**). It could be seen that the sensor array achieves good properties in these fields, indicating the broad application perspective in wearable electronics.

The main text and the Supplementary Note are revised according to your comment, which is listed as following for your convenience (all changes made in the revised MS is highlighted in yellow):

Main text, Table 1, Page 34

Table 1 Performance comparison with state-of-the-art pressure sensor

Sensing Mechanisms	Sensitivity/kPa	Detection range/kPa	Stretchability/%	Crosstalk isolation/dB	Studies
Contact resistance	18.94 (10 μm)	< 40	~ 50	~ 33.41	This work
Contact	4.04 (30 μm)	< 200	~ 40	--	Ref. 61
	4.88	0.37 ~ 8			

resistance					
Contact resistance	0.7	< 25	< 3	Low	Ref. 28
Contact resistance	~15.1	0.2 ~ 59	--	--	Ref. 31
Contact resistance	25.1	< 2.6	--	--	Ref. 32
Contact resistance	50.17	< 0.07	--	--	Ref. 58
Contact resistance	8.5	< 12	--	--	Ref. 29
Contact resistance	99.5	0.09 ~ 1	< 3	Low	Ref. 60
Resistance	15.22	< 5	< 3	--	Ref. 22
Resistance	0.0835	0.098 ~ 50	--	--	Ref. 8
Resistance	0.048	< 18	> 50	--	Ref. 30
Resistance	0.011	1 ~ 120	> 3	24.8	Ref. 15
Resistance	8.2	< 10	> 3	Obvious	Ref. 9
Resistance	~ 2.75	< 4	> 12	Low	Ref. 52
Resistance	~ 2.07	< 15	> 3	Obvious	Ref. 44
Resistance	--	--	> 3	Low	Ref. 53
Capacitive	~ 0.02	< 22	> 3	Low	Ref. 54
Capacitive	0.021	< 600	--	--	Ref. 57
Capacitive	~ 0.1	< 10	> 3	--	Ref. 14
Capacitive	44.5	< 0.1	> 3	--	Ref. 59
Capacitive	0.00023	< 800	> 100	Obvious	Ref. 43
Capacitive	~ 13	0.5 ~ 5	> 3	--	Ref. 65
Capacitive	0.55	0.2 ~ 7	> 3	Obvious	Ref. 42
Triboelectric	~ 1.5	0.2 ~ 3.65	> 3	Obvious	Ref. 55
Triboelectric	~ 0.45	< 20	--	--	Ref. 56
Triboelectric	~ 44.14	< 0.75	--	--	Ref. 62
Triboelectric	~ 0.046	< 170	--	--	Ref. 64
Triboelectric	~ 0.011	0.07 ~ 40	--	--	Ref. 38
Piezoelectric	~ 0.8	0.1 ~ 20.3	--	--	Ref. 63

Supplementary Note 8, Supplementary Figure 29, Page S39

Supplementary Figure 29 | Performance comparison with state-of-the-art pressure sensors.

Comparison of our sensor with existing sensors in terms of sensitivity and detection range are depicted in Supplementary Fig. 29¹⁻²⁶. Although it is not the most sensitive, it shows a wider linear response range. The two representative works are selected to further analyze the advantages of our sensor (shown on right), including stretchability, thickness, sensitivity, transparency, detection range and crosstalk isolation. It could be seen that our sensor has good performance in all aspects, which can be attached to human skin with large scale for precise tensile detection.

2. How much the thickness of the grid walls impact in the performances of the sensor (sensing range) and mechanical crosstalk should be better highlighted. Are there any sort of design rules that have to be applied and followed to reproducibly fabricate such systems? Some hints are given in the main text and, again, more details are given in the

SI, but I strongly suggest the authors to better highlight this point and give more details for the reader in the main text.

Response:

Thanks the reviewer for the suggestion. It is essential to explain the principle of isolating mechanical crosstalk in the main text, which is helpful for readers to understand the key points of this work. Therefore, we have fully modified **Fig. 1**, which is mainly used to elaborate the design concept of the sensor based on micro-cage structure. The structure of the sensor array and the characterization of *prs*/PDMS are shown in **Fig. 2**. **Figure 1a** illustrates the human sense of touch deals with spatiotemporal perception under external stimuli through many mechanoreceptors, and the *prs*/PDMS layer introduced here mimics the structure of human skin. Then a simple two-dimensional model is proposed to analyze the influence of different *prs*/PDMS designs on crosstalk isolation, including the four important parameters pixel length (l_p), spacer length (l_s) and thickness (t_s), and external displacement (D) (**Fig. 1c**). The simulation results show that the deformation overflow of devices using *prs*/PDMS layer is reduced by 90.3% compared to that of conventional flexible electronics. The crosstalk isolation could exceed 25 dB with the ratio of spacer length to pixel length being 1: 2, which seems to us to achieve the better isolation effect. In practical applications, an excessively long spacer is not conducive to fabricate array devices. So if the higher density devices need to be prepared and crosstalk isolation requirements are not very strict, this ratio could also be appropriately increased, such as 1: 5, or even 1: 10, etc. More detailed comparisons of crosstalk isolation are depicted in **Supplementary Table 2** and **Supplementary Table 3**.

The main text and the Supplementary Note are revised according to your comment, which is listed as following for your convenience (all changes made in the revised

MS is highlighted in yellow):

Main text, Figure 1, Page 29

Fig. 1 | Principle of ultralow crosstalk sensor with micro-cage structure based on *prs*/PDMS layer. **a** Schematic illustration of various independent mechanoreceptors in glabrous skin for accurate tactile perception. **b** Proposed chemistry of benzophenone inhibiting PDMS crosslinking under UV light to prepare *prs*/PDMS layer. **c** Simple geometric analysis of micro-cage structure formed by *prs*/PDMS under external pressure. **d** Two-dimensional deformation simulation analysis of the PDMS & *prs*/PDMS and PET under external pressure, and the micro-cage structure of the former enables the small elongation strain compared with the latter. **e** Quantitative analysis of displacement variation along the x-axis for different models, and the displacement of adjacent pixel using *prs*/PDMS layer decrease by 90.3% compared with PET. **f** Crosstalk isolation versus the ratio of spacer length to pixel length, the better isolation effect could be achieved when its value is greater than 1 : 2.

Main text, Page 6-7

Design concept of micro-cage structure based on *prs*/PDMS layer. Human sense of touch deals with spatiotemporal perception under external stimuli through a large number of receptors (Fig. 1a). The relevant information reaches the spinal cord through multiple nerves and is transmitted to the central nervous system via two main pathways for higher-level processing and interpretation: spinothalamic and dorsal-column-medial-lemniscal. The latter could quickly convey pressure/vibration information to the brain for precise tactile detection. Taking fingertips as an example, there are many mechanoreceptors embedded in skin at different depths, which are mainly divided into

four categories: slow-adapting receptors responding to static pressures (SA-I and SA-II) and fast-adapting receptors responding to dynamic forces or vibration (FA-I and FA-II). Some of these mechanoreceptors are distributed independently at the intermediate ridges between the epidermis and dermis, which could produce stress concentration to enhance pressure perception. Our pressure sensors mimic this structure and individual receptors, enabling each sensor to operate with ultralow crosstalk. As shown in Fig. 1b, doping benzophenone into PDMS will inhibit its crosslinking when exposed to UV light, thus forming the photo-reticulated PDMS (Supplementary Fig. 1). The micro-cage structure is formed after encapsulating with another layer of electrodes, and pressure sensor could be prepared within the cage. Besides, the boundary of the micro-cage composed by photo-reticulated PDMS could separate different sensors and prevent inevitable deformation diffusion, showing the effect of strain local confinement, so it is called photo-reticulated strain localization PDMS (*prs*/PDMS). A simple two-dimensional model is proposed to analyze the deformation of micro-cage structure under external pressure (Fig. 1c). It could be found that the entire model has four key parameters, which are pixel length (l_p), spacer length (l_s) and thickness (t_s), and external displacement (D). When a pixel is subjected to external pressure, both the top electrode and *prs*/PDMS spacer will deform, thereby gradually expanding to adjacent pixel, resulting in mechanical crosstalk. Deformation simulation analysis is performed on PDMS model with *prs*/PDMS spacer and the model only with PET (Fig. 1d). The results demonstrate that the former realizes the small elongation strain due to the strain confinement of *prs*/PDMS spacer, but PET substrate with high toughness and non-stretchable characteristics shows the large deformation. Moreover, a coordinate system is established on the right boundary of stressed pixel to quantitative analyze the elongation strain along the x-axis and y-axis (Fig. 1e). The displacement along the y-axis declines slowly for the PET model, and its average displacement in adjacent pixel is 6.13 μm . However, the deformation of *prs*/PDMS spacer model decreases rapidly in the spacer layer region, and the average displacement of adjacent pixel is only 0.56 μm , which decreases by 90.3% compared with PET substrate (the maximum deformation of stressed pixel). Figure 1f describes the crosstalk isolation versus the different ratios of spacer length to pixel length, and its value reaches 25.03 dB with the ratio of 0.5, which can be considered as better mechanical crosstalk isolation (More detailed analysis could refer to the Supplementary Fig. 10).

Supplementary Note 4, Page S16-17

Nevertheless, an excessively long spacer is not conducive to design the array device, so it is considered that the I_{so} above 25 dB has sufficient crosstalk isolation effect, and the ratio of spacer length to pixel length at this time is 1: 2. More detailed comparison data could refer to Supplementary Table 2 and Supplementary Table 3. In practical applications, if the higher density devices need to be prepared and crosstalk isolation requirements are not very strict, this ratio could also be appropriately increased, such as 1: 5, or even 1: 10, etc.

Supplementary Table 2 Crosstalk isolation between PET and *prs*/PDMS

	Displacement/ μm	Displacement of adjacent pixel/ μm	Crosstalk isolation/dB
PDMS & pr s/PDMS	2	0.19494	20.22258
	4	0.45338	18.91195
	6	0.43939	22.70602
	8	0.52455	23.66606
	10	0.58465	24.66208
PET	2	1.20711	4.38566
	4	2.3246	4.71424
	6	3.66558	4.28017
	8	4.81495	4.40996
	10	6.13427	4.24474

Supplementary Table 3 Crosstalk isolation of *pr*s/PDMS at different parameters

l_p : l_s : t_s / μm	Displacement/ μm	Displacement of adjacent pixel/ μm	Crosstalk isolation/dB
50: 10: 10	Max (10)	0.75512	22.43968
50: 15: 10		0.66014	23.60728
50: 20: 10		0.64746	23.77574
50: 25: 10		0.55772	25.07168
50: 30: 10		0.51086	25.83396
50: 25: 15	10	0.75512	25.02554
50: 25: 20	10	0.66014	26.39964
50: 25: 25	10	0.64746	26.82432

3. How much the reported procedure can be upscaled at low costs, for the routinely fabrication of such sensing systems over large areas?

Response:

Thanks the reviewer for the suggestion. It is essential to explore the capability of the device for the large-scale production, so we have made the following attempts:

1. A larger acrylic frame is used to collect the polyvinyl alcohol nanofibers (PVA NFs), whose diameter is about 6 cm, and the film with good uniformity could be obtained. Then the roller is further used to collect the nanofibers, which can be peeled off from the aluminum foil, and it still shows good transparency and tensile property. This method greatly improves the efficiency of nanofiber preparation, and it is suitable for the large-scale production (Fig. R1a).

2. The method to fabricate graphene films in this work can also be used for large area production. The ethanol-assisted graphene dispersion solution is evenly sprayed on the water surface, and the sponges are put on one side of the interface to quickly siphon water from the system, thereby forming a self-assembled graphene film at the liquid/air interface, as shown in the Fig. R1b. More detailed preparation process could refer to

the **Supplementary Movie 1**.

3. The *prs*/PDMS solution in this work shows the similar properties to photoresist, which is sensitive to ultraviolet light. It could be spin-coated on the desired substrates and patterned by conventional photolithography process. To sum up, our preparation process shows the advantages of simple operation, low cost and high efficiency, which is suitable for large-scale preparation of pressure sensors.

Figure R1 | Preparation of large-area nanofibers (a) and graphene films (b).

The main text and the Supplementary Note are revised according to your comment, which is listed as following for your convenience (all changes made in the revised MS is highlighted in yellow):

Main text, Page 8

The results show the *prs*/PDMS layer possesses the similar stretchability ($\sim 5.08\text{MPa}$) and transmittance (91.85% in visible light) with PDMS, allowing the preparation of transparent stretchable devices. For more detailed fabrication process, please refer to the Method, Supplementary Note 1 – 3 and Supplementary Movie 1.

4. The Ag electrodes configuration should be better explained, so far as I understood there is a common grounded electrodes and each pixel is measuring the current flowing between the second electrode and the common ground, is this correct?

Response:

Thanks the reviewer for the suggestion. As you have pointed out that one end of each pixel in sensor array is gathered together to form a common port and connected to the power supply. The other end of each pixel is connected to an external resistor and then grounded. This allows each pixel in series with resistor, while different pixels are connected in parallel. According to the Kirchhoff's law, the current in the loop is equal everywhere, so the current flowing through the external resistor can be considered as the current of the sensor. When the pressure on the sensor increases, its resistance will gradually decline. And then more current will be generated in the loop, which will increase the voltage of the external resistor. The multi-channel data acquisition card (PXIe-4300, National Instruments) is used to collect the voltage on external resistors, and then the pressure on sensor could be deduced, thus reflecting the real-time pressure distribution of the sensor array.

The main text and the Supplementary Note are revised according to your comment, which is listed as following for your convenience (all changes made in the revised MS is highlighted in yellow):

Supplementary Note 8, Supplementary Figure 22, Page S33

Supplementary Figure 22 | Multi-channel data measurement system. (a) Schematic illustration of the pressure mapping process (left). Optical image of the data measurement system (right). (b) The principle of the multi-channel synchronous data acquisition.

Supplementary Figure 22a demonstrates the schematic diagram of a 6×6 sensor array for pressure detection. The electrical signal of the pixels will change with the objects placed on its surface, thereby real-time presenting the shape of the object. The measurement principle here is similar to that of the previous multiple sensors, and each pixel in the array is connected to a resistor. Multi-channel synchronous data acquisition card collects the voltage on the resistor, then calculates the pressure on the pixel, which is equivalent to multi-electrometer scanning simultaneously.

Supplementary Note 7, Supplementary Figure 19, Page S29-30

Supplementary Figure 19 | The conversion principle of current and voltage in multi-channel data acquisition. (a) Equivalent circuit diagram of single-channel signal acquisition. Fitted curves of current (b) and resistance (c) of the sensor in the linear region of 20 to 200 kPa. Current variation (d) and the voltage variation on external resistance (e) with different resistors in series to the loop.

Supplementary Figure 19a depicts the equivalent circuit of single-channel signal acquisition. Based on the Kirchhoff's law, the current of the sensor could be obtained by the following formula:

$$I_s = I_r = \frac{U_r}{R_r} \quad (7-1)$$

I_s : the current of pressure sensor, I_r : the current of external resistance, U_r : the voltage of external resistance; Furthermore, the voltage of external resistance could also be deduced as follows:

$$U_r = \frac{R_r}{R_s + R_r} U_s \quad (7-2)$$

U_s : the voltage of power supply;

According to previous measurement data, the empirical formula of the current on sensor with the pressure could be fitted (Supplementary Fig. 19b), then the relationship between its resistance and pressure could also be calculated (Supplementary Fig. 19c):

$$I_s = 3.9 \times 10^{-9} P - 4.0 \times 10^{-5} \quad (7-3)$$

$$R_s = 89921.3 \exp\left(\frac{-P}{19196.0}\right) + 1913.4 \quad (7-4)$$

Subsequently, the influence of different external resistors on loop current and measurement voltage U_r is further studied. It could be found that the loop current gradually declines with the larger external resistance, but the range of U_r increases slightly which is beneficial for signal acquisition.

5. Ag printed electrodes generally gets oxidized upon continuous exposure to air. The authors should address this point and demonstrated that the system behavior is stable and reproducible over time and has a sufficient life time for the envisaged applications. In fact, change of resistivity of the bottom electrodes will affect the overall pixel resistance, according to the scheme reported in the SI.

Response:

Thanks the reviewer for the suggestion. The *prs*/PDMS layer in this work shows the function of adhesion and support, which could be used for packaging devices. Its surface will swell slightly during developing in toluene. After stacking the swollen layer with upper electrodes and continuously applying certain pressure, a new crosslinking network is formed in topological entanglement with preformed polymer networks, thus tightly adhering the two different layers. The interfacial toughness is measured about 55.94 J/m^2 , indicating that the electrode interfaces achieve the sufficient adhesion and can isolate external interference (**Supplementary Fig. 13d**). Therefore, Ag NFs electrodes won't be oxidized, and the sensitivity curve of the sensor is similar to the initial state even after exposure to ambient environment for 10 days.

The main text and the Supplementary Note are revised according to your comment, which is listed as following for your convenience (all changes made in the revised MS is highlighted in yellow):

Supplementary Note 5, Supplementary Figure 13, Page S21-22

Supplementary Figure 13 | The working principle and stability of the device. (a) Schematic illustration of the measurement setup. **(b)** Working principle of the pressure sensor. **(c)** Principle of robust interfacial adhesion between the top and bottom electrodes. **(d)** Characterization of interfacial toughness of device. **(e)** Performance comparison of device exposed to air for 10 days. Inset: Sensitivity comparison of devices in different regions.

The pressure sensor is composed of microstructured graphene layer, *prs*/PDMS layer and Ag NFs interdigital electrodes, as shown in Supplementary Fig. 13a. A circuit model has been proposed to analyze the resistance variation of the sensor under external

pressure (Supplementary Fig. 13b). It could be found that the resistance of the sensor is affected by three aspects, including the resistance of graphene (R_{Gr}), the resistance of Ag NFs (R_{Ag}) and the contact resistance between two electrodes (R_C). Hence, the current of sensor could be calculated by the following equation:

$$I = U / (R_{Ag} + R_C + R_{Gr}) \quad (5-1)$$

The R_{Ag} and R_{Gr} are the constant value which could be regulated by improving the preparation process. Therefore, the contact resistance (R_C) is the main factor affecting the conductance of the sensor, which mainly depends on the contact area. The small contact area at a low pressure shows a large contact resistance, and the contact area increasing rapidly with the external pressure, leading to an increase in conductivity of the sensor. Supplementary Figure 13c describes the principle of interfacial adhesion between the top and bottom layers. The *pr*s/PDMS layer is swollen in toluene during its preparation. After stacking the swollen layer with the microstructure layer and continuously applying certain pressure, monomers and crosslinkers will infiltrate into the preformed PDMS networks. Then, a new crosslinking network is formed in topological entanglement with preformed polymer networks, thus tightly adhering the two different layers. The interfacial toughness was measured about 55.94 J/m², indicating that the electrode interfaces achieve the sufficient adhesion and can isolate external interference (Supplementary Fig. 13d). Therefore, the device shows an excellent performance, and its pressure detection ability is similar to the initial state even after exposure to ambient environment for 10 days (Supplementary Fig. 13e).

6. NO statistical analysis has been reported, please add details about it. How many devices and arrays have been measured? The authors must report all the graphs with the error bars, same for the reported sensitivities.

Response:

Thanks the reviewer for the suggestion. As the reviewer suggested, three to five devices of each type have been prepared to observe their performance variation. The figures in the manuscript have been modified after statistically analyzing the data, the results show that our sensor achieve the stable performance.

The main text and the Supplementary Note are revised according to your comment, which is listed as following for your convenience (all changes made in the revised MS is highlighted in yellow):

Main text, Page 31

Fig. 3 | Pressure sensing performance of the ultrathin sensor. **a** Schematic diagram of the experimental setup for pressure sensing. **b** Sensitivity curve of the sensor (top); Simulation of stress distribution of the *pr*s/PDMS layer and microstructured PDMS under external pressure (middle); Cross-sectional SEM images of the sensor under different pressures (bottom). Relative change in current ($\Delta I/I_0$) of the sensors with different interdigital Ag NFs electrodes (c), spacer layer thickness (d) and pyramid sizes (e). Relative change in current ($\Delta I/I_0$) of the sensor with different strains (f) and bending radii (g). Inset: sensor response after recovery from deformations (left) and to tiny forces even in the bending state (right). **h** Stability measurement of the pressure sensor, and the cycle period is over 5000 cycles. Inset: The response performance of the sensor with different spacer thickness.

Supplementary Note 5, Supplementary Figure 14, Page S23

Supplementary Figure 14 | Sensitivity analysis of the sensors with different interdigital electrodes (a), spacer layer thickness (b) and pyramid sizes (c).

Supplementary Note 5, Supplementary Figure 15, Page S24

Supplementary Figure 15 | Sensitivity (a) and detection limit (b) of sensors under different deformations.

7. I understand that bending of the substrate is not affecting the system sensitivity, which is a very important point. But I guess that transferring such system in a rough, uneven, substrate could dramatically change the sensitivity from pixel to pixel, how can the authors deal with this issue?

Response:

Thanks the reviewer for the suggestion. As the reviewer suggested, it is essential to investigate the sensitivity variation of devices on rough or uneven substrates. The *prs*/PDMS layer shows the good adhesion and support, so it could provide air gap for pressure sensing even under bending conditions. A smaller bending radius means the smaller gap in the sensor, and the corresponding initial current will increase. On this basis, the linear detection range will decline, but its sensitivity does not change significantly. The experiments are consistent with the theoretical analysis, its linear range at the bending radius of 9.57 mm extends to 100 kPa, while the sensor with the radius of 4.60 mm reaches saturation after 50 kPa, but both show the similar sensitivity. Therefore, our pressure sensor has the ability to detect pressure on the uneven substrate.

The main text and the Supplementary Note are revised according to your comment, which is listed as following for your convenience (all changes made in the revised MS is highlighted in yellow):

Supplementary Note 5, Supplementary Figure 16, Page S25

Supplementary Figure 16 | Pressure response performance of the sensor under bending condition.

Supplementary Figure 16a describes the schematic diagram of the device under bending condition, in which *prs*/PDMS layer has a good adhesion effect on the upper and bottom layers, so there is still an air gap in sensor. However, the air gap decreased with the smaller bending radius, which also means the sensor is in the linear response region. The currents in two different bending states are illustrated in Supplementary Fig. 16b, and the smaller radius shows the larger current, indicating a closer contact between electrodes. The pressure response performance of the sensor under bending conditions was further explored. The results show that its sensitivity in the linear region is basically the same, but the device with smaller bending radius presents lesser pressure detection range, which quickly enters the saturation region. For the sensor with bending radius of 9.57 mm, the linear pressure response range extends to 100 kPa (Supplementary Fig. 16c), and this result is consistent with theoretical analysis.

8. English must be strongly polished as there are several mistakes throughout the whole text.

Response:

Thanks the reviewer for the suggestion. We reviewed the paper carefully and corrected the mistakes in it.

The main text and the Supplementary Note are revised according to your comment, which is listed as following for your convenience (all changes made in the revised MS is highlighted in yellow):

Main text, Page 7

1. Moreover, a coordinate system is established on the right boundary of stressed pixel to **quantitative analyze** the elongation strain along the x-axis and y-axis (Fig. 1e).

Main text Page 9

2. Extensive experimental data (Fig. 3b) show that the **sensitivity curve** could be divided into three regions.

3. Therefore, although the **sensitivity curves** shift to the right.

Main text, Page 31

4. **Sensitivity curve** of the sensor (top);

Reviewer #4:

The authors propose a tactile sensor array with encapsulated patterning that is aimed at enhancing the spatial selectivity of the device. This design, in the aims of the authors, would reduce the cross-talk among adjacent taxels and thus improve the ability to localize tactile stimuli that are spatially distributed.

Overall, the proposed technology is sound and it deserves attention, however I would recommend addressing some issues as detailed hereafter.

Answers:

We would like to express our sincere thanks to the referee for her/his great effort to review the manuscript and positive evaluation on our work.

1. First of all, the narrative grounded on the major need, in tactile sensing, to have transducers not affected by cross-talk is questionable. There are several reports, both in somatosensory neurophysiology and in biomimetic artificial touch studies, that point out that cross talk among adjacent sensors may be used as a tool to localize the stimulus thanks to triangulation mechanisms (and machine learning methods in some recent studies). Therefore, I would recommend revising the narrative of the paper and taking into account that cross-talk may also be beneficial to enhance the ability to localize the stimulus.

Response:

Thanks the reviewer for the suggestion. We have reviewed the relevant literature and agree with your opinion. In the field of somatosensory neurophysiology, crosstalk can indeed enhance the local stimulus, thus further amplifying the external signal and making the body respond to it. However, for the multi-point stimulation, if the stress point shows the large crosstalk to its surrounding, it will be difficult to distinguish the stress point, which is not conducive to accurate tactile perception (**Fig. 5**). The mechanoreceptors in human skin are independent of each other, which can detect external stimuli independently, and can also work together for a large area tactile perception. The *prsl*/PDMS layer introduced here mimics the above structure, which acquires the high transparency and stretchability. After separating the different pixels, the device can not only efficiently differentiate the stress point, but also detect the pressure distribution imaging, which will have a profound impact on future tactile sensors.

To sum up, we have fully modified the **Fig. 1** to help readers to understand the key points of this work. The structure of the sensor array and the characterization of *prsl*/PDMS are now represented in **Fig. 2**. The *prsl*/PDMS layer could adhere the top electrode and bottom electrode to form a micro-cage structure, and reduce the inevitable deformation diffusion. Then a simple two-dimensional model is proposed to analyze the influence of different *prsl*/PDMS designs on crosstalk isolation, including the four important parameters pixel length (l_p), spacer length (l_s) and thickness (t_s), and external displacement (D) (**Fig. 1c**). The simulation results show that the deformation overflow

of devices using *prs*/PDMS layer is reduced by 90.3% compared to that of conventional flexible electronics. Finally, the influence of *prs*/PDMS on crosstalk isolation with various designs is analyzed. It could exceed 25 dB with the ratio of spacer length to pixel length being 1: 2, which seems to us to achieve the better isolation effect. In practical applications, an excessively long spacer is not conducive to fabricate array devices. So if the higher density devices need to be prepared and crosstalk isolation requirements are not very strict, this ratio could also be appropriately increased, such as 1: 5, or even 1: 10, etc. More detailed comparisons of crosstalk isolation are depicted in **Supplementary Table 2** and **Supplementary Table 3**.

The main text and the Supplementary Note are revised according to your comment, which is listed as following for your convenience (all changes made in the revised MS is highlighted in yellow):

Main text, Figure 1, Page 29

Fig. 1 | Principle of ultralow crosstalk sensor with micro-cage structure based on *prs*/PDMS layer. **a** Schematic illustration of various independent mechanoreceptors in glabrous skin for accurate tactile perception. **b** Proposed chemistry of benzophenone inhibiting PDMS crosslinking under UV light to prepare *prs*/PDMS layer. **c** Simple geometric analysis of micro-cage structure formed by *prs*/PDMS under external pressure. **d** Two-dimensional deformation simulation analysis of the PDMS &

prs/PDMS and PET under external pressure, and the micro-cage structure of the former enables the small elongation strain compared with the latter. **e** Quantitative analysis of displacement variation along the x-axis for different models, and the displacement of adjacent pixel using *prs*/PDMS layer decrease by 90.3% compared with PET. **f** Crosstalk isolation versus the ratio of spacer length to pixel length, the better isolation effect could be achieved when its value is greater than 1: 2.

Main text, Page 6-7

Design concept of micro-cage structure based on *prs*/PDMS layer. Human sense of touch deals with spatiotemporal perception under external stimuli through a large number of receptors (Fig. 1a). The relevant information reaches the spinal cord through multiple nerves and is transmitted to the central nervous system via two main pathways for higher-level processing and interpretation: spinothalamic and dorsal-column-medial-lemniscal. The latter could quickly convey pressure/vibration information to the brain for precise tactile detection. Taking fingertips as an example, there are many mechanoreceptors embedded in skin at different depths, which are mainly divided into four categories: slow-adapting receptors responding to static pressures (SA-I and SA-II) and fast-adapting receptors responding to dynamic forces or vibration (FA-I and FA-II). Some of these mechanoreceptors are distributed independently at the intermediate ridges between the epidermis and dermis, which could produce stress concentration to enhance pressure perception. Our pressure sensors mimic this structure and individual receptors, enabling each sensor to operate with ultralow crosstalk. As shown in Fig. 1b, doping benzophenone into PDMS will inhibit its crosslinking when exposed to UV light, thus forming the photo-reticulated PDMS (Supplementary Fig. 1). The micro-cage structure is formed after encapsulating with another layer of electrodes, and pressure sensor could be prepared within the cage. Besides, the boundary of the micro-cage composed by photo-reticulated PDMS could separate different sensors and prevent inevitable deformation diffusion, showing the effect of strain local confinement, so it is called photo-reticulated strain localization PDMS (*prs*/PDMS). A simple two-dimensional model is proposed to analyze the deformation of micro-cage structure under external pressure (Fig. 1c). It could be found that the entire model has four key parameters, which are pixel length (l_p), spacer length (l_s) and thickness (t_s), and external displacement (D). When a pixel is subjected to external pressure, both the top electrode and *prs*/PDMS spacer will deform, thereby gradually expanding to adjacent pixel, resulting in mechanical crosstalk. Deformation simulation analysis is performed on PDMS model with *prs*/PDMS spacer and the model only with PET (Fig. 1d). The results demonstrate that the former realizes the small elongation strain due to the strain confinement of *prs*/PDMS spacer, but PET substrate with high toughness and non-stretchable characteristics shows the large deformation. Moreover, a coordinate system is established on the right boundary of stressed pixel to quantitative analyze the elongation strain along the x-axis and y-axis (Fig. 1e). The displacement along the y-axis declines slowly for the PET model, and its average displacement in adjacent pixel is 6.13 μm . However, the deformation of *prs*/PDMS spacer model decreases rapidly in the spacer layer region, and the average displacement of adjacent pixel is only 0.56 μm , which decreases by 90.3% compared with PET substrate (the maximum deformation of

stressed pixel). Figure 1f describes the crosstalk isolation versus the different ratios of spacer length to pixel length, and its value reaches 25.03 dB with the ratio of 0.5, which can be considered as better mechanical crosstalk isolation (More detailed analysis could refer to the Supplementary Fig. 10).

2. Considering the sectioning, I would suggest moving to the methods the subsections on device fabrication that are currently reported in the results section.

Response:

Thanks the reviewer for the suggestion. As the reviewer suggested, we put the device preparation in the **Method** section, and add a new **Fig. 2** to depict the structure of the sensor array and the characterization of *prs*/PDMS.

The main text and the Supplementary Note are revised according to your comment, which is listed as following for your convenience (all changes made in the revised MS is highlighted in yellow):

Main text, Page 17-18

The *prs*/PDMS was then spin-coated on the desired substrate and exposed to UV radiation by using a portable UV lamp (14 mW/cm², 10min). Benzophenone radicals will be generated, which will react with the silicon hydride groups in PDMS crosslinkers and the vinyl groups of PDMS monomers, thus preventing the traditional crosslinking reactions. A soft bake procedure was performed in a convection oven at 120°C for approximately 150s, and the unexposed PDMS will cure during the post exposure baking, while the exposed PDMS remains uncrosslinked and could be washed away in toluene. The thickness of the *prs*/PDMS layer could be controlled by varying the spin speed or the dilution ratio, and the sample was rinsed in isopropanol and blown with N₂ gas.

Main text, Page 30

Fig. 2 | Structure and design concept of the pressure sensor. **a** Schematic illustration of the structure of pressure sensor arrays. **b** SEM images of *prs*/PDMS layer with different resolutions, which could reach 100 μm ; The right photographs demonstrate the patterns on various substrates (top: glass, bottom: silicon wafer), indicating its good adaptability. **c** Tensile properties (left) and UV-visible spectra (right) of *prs*/PDMS exhibit the excellent stretchability and transparency. **d** Schematic showing the design concept for the ultralow crosstalk sensor, where the *prs*/PDMS layer perform the effect of strain local confinement for mechanical stability as well as multilayer device. Optical photographs of sensor arrays attached on palm no matter in the flat (**e**) or curly (**f**) state, which achieve the transparency of 50.36 % and thickness of 60.59 μm .

Main text, Page 7-8

Structure of pressure sensor array. Figure 2a shows the exploded view of the stretchable ultralow crosstalk sensor array. The device mainly consists of three parts: the patterned Ag NFs interdigital electrodes, the patterned *prs*/PDMS layer and graphene attached to the PDMS with pyramid microstructures. The main challenge in this work is to prepare the patterned dielectric films (*prs*/PDMS), which can not only separate adjacent pixels to form micro-cage structure, but also show the function of adhesion and support for packaging devices. SEM images of *prs*/PDMS layer with different resolutions are shown in Fig. 2b, with a precision up to 100 μm . Then the patterning effect on different substrates was also verified, such as glass (horse) and silicon (office logo), indicating its excellent adaptability. In addition, the stretchability and transmittance of *prs*/PDMS layer were further demonstrated in Fig. 2c. The results show the *prs*/PDMS layer possesses the similar stretchability ($\sim 5.08\text{MPa}$) and transmittance (91.85% in visible light) with PDMS, allowing the preparation of transparent stretchable devices. For more detailed fabrication process, please refer to

the Method, Supplementary Note 1 – 3 and Supplementary Movie 1. Since the high precision transparent stretchable *pr*s/PDMS film could achieve strain local confinement, it can be used to prepare multilayer devices and improve the mechanical stability, which could provide a solid foundation for more sophisticated electronics (Fig. 2d). Figure 2e and 2f show the 6 × 6 sensor array is well attached on the palm no matter in the flat or curly state. The insert figure (Fig. 2e) exhibits the single sensor on fingertip (2.0 × 2.0 mm²) with a transmittance of 50.36%, and the enlarged cross-section SEM image (Fig. 2f) shows the thickness of multilayer stacked structure is only 60.59 μm.

3. I would also recommend introducing a quantitative comparative discussion of results with respect to the metrological characteristics of pertinent state of the art sensors. Moreover, statistical analysis of experimental data should be added together with a presentation of the experimental protocols undertaken (including description of stimuli and strategy of stimuli administration, number of repetitions, and statistical indicators).

Response:

Thanks the reviewer for the suggestion. As the reviewer suggested, the performance comparison with other existing sensors is illustrated in **Table 1**. Furthermore, two representative works are selected to reflect the advantages of our sensor, including stretchability, thickness, sensitivity, transparency, detection range and crosstalk isolation (**Supplementary Fig. 29**). It could be seen that the sensor array achieves good properties in these fields, indicating the broad application perspective in wearable electronics. Moreover, three to five devices of each type have been prepared to observe their performance variation. The figures in the manuscript have been modified after statistically analyzing the data, the results show that our sensor achieve the stable performance.

The main text and the Supplementary Note are revised according to your comment, which is listed as following for your convenience (all changes made in the revised MS is highlighted in yellow):

Main text, Table 1, Page 34

Table 1 Performance comparison with state-of-the-art pressure sensor

Sensing Mechanisms	Sensitivity/kPa	Detection range/kPa	Stretchability/%	Crosstalk isolation/dB	Studies
Contact resistance	18.94 (10 μm)	< 40	~ 50	~ 33.41	This work
Contact resistance	4.04 (30 μm)	< 200	~ 40	--	Ref. 61
Contact resistance	4.88	0.37 ~ 8	~ 40	--	Ref. 28
Contact resistance	0.7	< 25	< 3	Low	Ref. 28
Contact	~ 15.1	0.2 ~ 59	--	--	Ref. 31

resistance					
Contact resistance	25.1	< 2.6	--	--	Ref. 32
Contact resistance	50.17	< 0.07	--	--	Ref.58
Contact resistance	8.5	< 12	--	--	Ref. 29
Contact resistance	99.5	0.09 ~ 1	< 3	Low	Ref. 60
Resistance	15.22	< 5	< 3	--	Ref. 22
Resistance	0.0835	0.098 ~ 50	< 50	--	Ref. 8
Resistance	0.048	< 18	< 50	--	Ref. 30
Resistance	0.011	1 ~ 120	< 3	24.8	Ref. 15
Resistance	8.2	< 10	< 3	Obvious	Ref. 9
Resistance	~ 2.75	< 4	< 12	Low	Ref. 52
Resistance	~ 2.07	< 15	< 3	Obvious	Ref. 44
Resistance	--	< 22	< 3	Low	Ref. 53
Capacitive	~ 0.02	< 22	< 3	Low	Ref. 54
Capacitive	0.021	< 600	< 3	--	Ref. 57
Capacitive	~ 0.1	< 10	< 3	--	Ref. 14
Capacitive	44.5	< 0.1	< 3	--	Ref. 59
Capacitive	0.00023	< 800	< 3	Obvious	Ref. 43
Capacitive	~ 13	0.5 ~ 5	< 3	--	Ref. 65
Capacitive	0.55	0.2 ~ 7	< 3	Obvious	Ref. 42
Triboelectric	~ 1.5	0.2 ~ 3.65	< 3	Obvious	Ref. 55
Triboelectric	~ 0.45	< 20	--	--	Ref. 56
Triboelectric	~ 44.14	< 0.75	--	--	Ref. 62
Triboelectric	~ 0.046	< 170	--	--	Ref. 64
Triboelectric	~ 0.011	0.07 ~ 40	--	--	Ref. 38
Piezoelectric	~ 0.8	0.1 ~ 20.3	--	--	Ref. 63

Main text, Page 14

Table 1 illustrates the performance comparison with state-of-the-art pressure sensor, including sensitivity, detection range, stretchability and crosstalk isolation⁵⁴⁻⁶⁵, and the results reveal that the sensor array in this work achieve good properties in these fields (Supplementary Fig. 29).

Supplementary Note 8, Supplementary Figure 29, Page S39

Supplementary Figure 29 | Performance comparison with state-of-the-art pressure sensors.

Comparison of our sensor with existing sensors in terms of sensitivity and detection range are depicted in Supplementary Fig. 29¹⁻²⁶. Although it is not the most sensitive, it shows a wider linear response range. The two representative works are selected to further analyze the advantages of our sensor (shown on right), including stretchability, thickness, sensitivity, transparency, detection range and crosstalk isolation. It could be seen that our sensor has good performance in all aspects, which can be attached to human skin with large scale for precise tensile detection.

Main text, Page 31

Fig. 3 | Pressure sensing performance of the ultrathin sensor. **a** Schematic diagram of the experimental setup for pressure sensing. **b** Sensitivity curve of the sensor (top); Simulation of stress distribution of the *pr*s/PDMS layer and microstructured PDMS under external pressure (middle); Cross-sectional SEM images of the sensor under different pressures (bottom). Relative change in current ($\Delta I/I_0$) of the sensors with different interdigital Ag NFs electrodes (**c**), spacer layer thickness (**d**) and pyramid sizes (**e**). Relative change in current ($\Delta I/I_0$) of the sensor with different strains (**f**) and bending radii (**g**). Inset: sensor response after recovery from deformations (left) and to tiny forces even in the bending state (right). **h** Stability measurement of the pressure sensor, and the cycle period is over 5000 cycles. Inset: The response performance of the sensor with different spacer thickness.

Supplementary Note 5, Supplementary Figure 14, Page S23

Supplementary Figure 14 | Sensitivity analysis of the sensors with different interdigital electrodes (a), spacer layer thickness (b) and pyramid sizes(c).

Supplementary Note 5, Supplementary Figure 15, Page S24

Supplementary Figure 15 | Sensitivity (a) and detection limit (b) of sensors under different deformations.

4. The supplementary materials are very good for the sake of reproducibility of the processes, however for the same purpose I would also suggest sharing experimental data and elaboration code in open manner by means of some kind of repository.

Response:

Thanks the reviewer for the suggestion. As the reviewer suggested, we have sorted out all the experimental data and put it in the **Source data**. The data acquisition program in this work is based on LabVIEW, and the corresponding code has been put in the

Supplementary Information.

The main text and the Supplementary Note are revised according to your comment, which is listed as following for your convenience (all changes made in the revised MS is highlighted in yellow):

Supplementary Note 8, Supplementary Figure 26-28, Page S37-38

Supplementary Figure 26 | LabVIEW program for single sensor to collect current.

Supplementary Figure 27 | LabVIEW program for multi-channel synchronous voltage acquisition.

Supplementary Figure 28 | LabVIEW program for human-computer interaction.

machine interaction game.

5. The authors often use high resolution (e.g., high spatial resolution, high pressure resolution, ...), however in metrological terms better performance is associated with lower resolution; please revise somehow.

Response:

Thanks the reviewer for the suggestion. As the reviewer suggested, we have used more pertinent words to revise the title and some expressions in the article.

The main text and the Supplementary Note are revised according to your comment, which is listed as following for your convenience (all changes made in the revised MS is highlighted in yellow):

Main text, Title

Localizing Strain via Micro-Cage Structure for Stretchable Pressure Sensor Arrays with Ultralow Spatial Crosstalk

Main text, Page 2

Hence, the sensor achieves the **sufficient pressure resolution** to detect 1g weight even in bending condition, and it could monitor human pulse under different states or analyze the grasping postures.

Main text, Page 10

The sensor could distinguish the tiny weights of 1g, 2g and 5g at the bending radius of about 5 mm, suggesting the **good pressure resolution**.

6. In the section about “Structure and fabrication of ultralow crosstalk sensor” the authors suddenly introduce the horse and office logo patterning, however the reason why this is done is not clear. Please clarify.

Response:

Thanks the reviewer for the suggestion. The *prs*/PDMS patterning was carried out on different substrates to demonstrate its adaptability. The horse pattern was on glass substrate, while the office logo was on silicon. As the reviewer suggested, we have modified the text and gave some explanations.

The main text and the Supplementary Note are revised according to your comment, which is listed as following for your convenience (all changes made in the revised MS is highlighted in yellow):

Main text, Page 8

Then the patterning effect on different substrates was also verified, such as glass (horse) and silicon (office logo), indicating its excellent adaptability. In addition, the stretchability and transmittance of *prs*/PDMS layer were further demonstrated in Fig. 2c.

7. To quantitative analysis the elongation strain  to quantitative analyse the elongation strain.

Response:

Thanks the reviewer for the suggestion. We reviewed the paper carefully and corrected the mistakes in it.

The main text and the Supplementary Note are revised according to your comment, which is listed as following for your convenience (all changes made in the revised MS is highlighted in yellow):

Main text, Page 7

Moreover, a coordinate system is established on the right boundary of stressed pixel to **quantitative analyze** the elongation strain along the x-axis and y-axis (Fig. 1e).

8. Sensing sensitivity is redundant: maybe you can just use sensitivity.

Response:

Thanks the reviewer for the suggestion. We reviewed the paper carefully and corrected these mistakes.

The main text and the Supplementary Note are revised according to your comment, which is listed as following for your convenience (all changes made in the revised MS is highlighted in yellow):

Main text Page 9

Extensive experimental data (Fig. 3b) show that the **sensitivity curve** could be divided into three regions.

Therefore, although the **sensitivity curves** shift to the right.

Main text, Page 31

Sensitivity curve of the sensor (top);

9. The sentence “Furthermore, the sensor was carried out by the repetitive compression ...” is not clear.

Response:

Thanks the reviewer for the suggestion. We have modified this sentence and used a more rigorous expression to describe its stability test.

The main text and the Supplementary Note are revised according to your comment, which is listed as following for your convenience (all changes made in the revised MS is highlighted in yellow):

Main text Page 11

The durability of the sensor was detected by a repetitive contact test for more than 5000 cycles and the relative change in current almost unchanged, exhibiting an excellent stability (Fig. 3h).

REVIEWER COMMENTS

Reviewer #2 (Remarks to the Author):

The paper is revised well.

Reviewer #3 (Remarks to the Author):

I have carefully read the revised version of the manuscript, and I think that the authors have addressed all the concerns/comments arised by the reviwers, therefore, I do consider th manuscrit acceptable for publication in Nature Communication in the present form.

Reviewer #4 (Remarks to the Author):

The authors carefully revised the manuscript, with several improvements associated to technological aspects, benchmarking with respect to at least part of state of the art, and data availability.

However, in the revised manuscript, the authors state that: "Some of these mechanoreceptors are distributed independently at the intermediate ridges between the epidermis and dermis, which could produce stress concentration to enhance pressure perception. Our pressure sensors mimic this structure and individual receptors, enabling each sensor to operate with ultralow crosstalk."

In the reviewer's opinion, this not correctly justifying biomimetism, because even type-1 receptors (SA1, FA1) have cross-talk among adjacent units. Indeed, fingerprints act as strain concentrators and amplifiers, however adjacent receptors have overlaps in terms of receptive field. This is confirmed by the indepentent observations of several scientists, as an example in the papers by Johansson (1978) and by Vega-Bermudez and Johnson (1999). Just to mention a quantitative figure, in the fingertips a 50 micrometers indentation results into a receptive field of about 5 mm² for both Merkel and Meissner receptors. Remarkably, in the fingertips there are about 70 Merkel units per mm² and about 140 Meissner per mm². This means that several units are activated in parallel, even in case of very gentle tactile experiences.

In light of these background findings, I would again recommend, with constructive attitude, a revision of the narrative of the paper. Of course to have artificial touch sensors with separated receptive fields could represent an added value in several field applications, however the authors cannot claim that this is biomimetic as confirmed by the mentioned studies (among several) of human touch neurophysiology.

I look forward to another revision round of this promising study.

Best regards,

Calogero Maria Oddo

Johansson RS: Tactile sensibility in the human hand: receptive field characteristics of mechanoreceptive units in the glabrous skin area. *J Physiol (Lond)* 1978, 281:101-123.

Vega-Bermudez F, Johnson KO: SA1 and RA receptive fields, response variability, and population responses mapped with a probe array. *J Neurophysiol* 1999, 81:2701-2710.

Point to Point Response to the referees' reports

(comments in black, responses in blue, changes highlighted in yellow):

Reviewer #2:

The paper is revised well.

Response:

We would like to express our sincere thanks to the referee for her/his great effort to review the manuscript.

Reviewer #3:

I have carefully read the revised version of the manuscript, and I think that the authors have addressed all the concerns/comments raised by the reviewers, therefore, I do consider the manuscript acceptable for publication in Nature Communication in the present form.

Response:

We would like to express our sincere thanks to the referee for her/his great effort to review the manuscript.

Reviewer #4:

The authors carefully revised the manuscript, with several improvements associated to technological aspects, benchmarking with respect to at least part of state of the art, and data availability.

However, in the revised manuscript, the authors state that: “Some of these mechanoreceptors are distributed independently at the intermediate ridges between the epidermis and dermis, which could produce stress concentration to enhance pressure perception. Our pressure sensors mimic this structure and individual receptors, enabling each sensor to operate with ultralow crosstalk.”

In the reviewer’s opinion, this not correctly justifying biomimetism, because even type-1 receptors (SA1, FA1) have cross-talk among adjacent units. Indeed, fingerprints act as strain concentrators and amplifiers, however adjacent receptors have overlaps in terms of receptive field. This is confirmed by the independent observations of several scientists, as an example in the papers by Johansson (1978) and by Vega-Bermudez and Johnson (1999). Just to mention a quantitative figure, in the fingertips a 50 micrometers indentation results into a receptive field of about 5 mm² for both Merkel and Meissner receptors. Remarkably, in the fingertips there are about 70 Merkel units per mm² and about 140 Meissner per mm². This means that several units are activated in parallel, even in case of very gentle tactile experiences.

In light of these background findings, I would again recommend, with constructive attitude, a revision of the narrative of the paper. Of course to have artificial touch sensors with separated receptive fields could represent an added value in several field applications, however the authors cannot claim that this is biomimetic as confirmed by the mentioned studies (among several) of human touch neurophysiology.

Response:

We would like to express our sincere thanks to the referee for your great effort to review the manuscript and positive evaluation on our work. I have carefully read the literature you listed and other related literature (Nat. Rev. Neurosci. 2009, 10, 345-359), as you said, mechanoreceptors on fingertips show the different receptive fields. When using the hand to grasp an object, the skin will conform to its surface and maintain the same local contour, thereby projecting deformation to a variety of mechanoreceptors. Each mechanoreceptor represents a small portion of the object and encodes the spatiotemporal tactile information as spikes of action potentials, which is transmitted to the central nervous system for higher level processing, such as precise touch and kinesthesia (eLife 2014, 3, e01488 and IEEE Trans. Robot. 2010, 26, 1-20). In this work, we try to solve the problem of mechanical crosstalk caused by intensive multi-point stimulation of artificial touch sensors. By introducing the *prs*/PDMS layer, different receptive fields could be separated and clear real-time pressure distribution imaging is achieved. This is another strategy to realize precise tactile perception, so we have revised the description in the article to make it easier for readers to understand. Thanks again to the reviewer for the corrections.

The main text and the Supplementary Note are revised according to your comment, which is listed as following for your convenience (all changes made in the revised MS is highlighted in yellow):

Main text, Figure 1, Page 29

Fig. 1 | Principle of ultralow crosstalk sensor with micro-cage structure based on *prs*/PDMS layer. **a** Schematic illustration of multiple mechanoreceptors in glabrous skin for accurate tactile perception. **b** Proposed chemistry of benzophenone inhibiting PDMS crosslinking under UV light to prepare *prs*/PDMS layer. **c** Simple geometric analysis of micro-cage structure formed by *prs*/PDMS under external pressure. **d** Two-dimensional deformation simulation analysis of the PDMS & *prs*/PDMS and PET under external pressure, and the micro-cage structure of the former enables the small elongation strain compared with the latter. **e** Quantitative analysis of displacement variation along the x-axis for different models, and the displacement of adjacent pixel using *prs*/PDMS layer decrease by 90.3% compared with PET. **f** Crosstalk isolation versus the ratio of spacer length to pixel length, the better isolation effect could be achieved when its value is greater than 1: 2.

Main text, Page 6-7

Taking fingertips as an example, there are many mechanoreceptors embedded in skin at different depths, which are mainly divided into four categories: slow-adapting receptors responding to static pressures (SA-I and SA-II) and fast-adapting receptors

responding to dynamic forces or vibration (FA-I and FA-II). These mechanoreceptors show the different receptive fields, and they work together to achieve the precise tactile perception. Our pressure sensor adopts another strategy to achieve the above effect, which is to separate the receptive field by introducing a dielectric layer, thereby preparing the tactile sensor with ultralow crosstalk. As shown in Fig. 1b, doping benzophenone into PDMS will inhibit its crosslinking when exposed to UV light, thus forming the photo-reticulated PDMS (Supplementary Fig. 1).

REVIEWER COMMENTS

Reviewer #4 (Remarks to the Author):

Dear Authors,

the paper is progressively improving during the review process, and the discussion of touch physiology is now better grounded thanks to the awareness that cross-talk among adjacent receptors is part of the natural encoding of tactile experience in animals.

However, there are still some sentences to be smoothed, to avoid over-stressing the strict need to limit cross-talk, which is not necessarily negative as shown by several studies about touch neurophysiology and artificial tactile sensors.

As an example, the authors state in the abstract that "However, denser arrays mean greater mechanical crosstalk among adjacent pixels, which significantly impair the detection accuracy". This is not supported by evidence and state of the art opinions in my judgment. Moreover, in the introduction the authors state that "but the crosstalk among adjacent pixels will significantly affect its spatial resolution and practical applications 9-16".

However, in my opinion, references 9 to 16 are not supporting such sentence.

I would suggest further smoothing the narrative through the whole paper (not just in these parts that I am mentioning here): you may claim the development of a sensor without cross-talk, but please to not associate this to biomimetism or to a strict technological requirement. You can as an example associate your design choice to ease of sensor calibration, but other solutions (with cross-talk) could show other benefits (redundancy, resiliency to disturbance, fault tolerance, ...).

Please consider my comments as constructive notes to contribute improving the paper, which has added value in my opinion and deserves attention for publication as soon as these aspects will be properly framed.

Best regards,

Calogero Maria Oddo

Point to Point Response to the referees' reports

(comments in black, responses in blue, changes highlighted in yellow):

Reviewer #4:

Dear Authors,

the paper is progressively improving during the review process, and the discussion of touch physiology is now better grounded thanks to the awareness that cross-talk among adjacent receptors is part of the natural encoding of tactile experience in animals.

However, there are still some sentences to be smoothed, to avoid over-stressing the strict need to limit cross-talk, which is not necessarily negative as shown by several studies about touch neurophysiology and artificial tactile sensors.

As an example, the authors state in the abstract that "However, denser arrays mean greater mechanical crosstalk among adjacent pixels, which significantly impair the detection accuracy". This is not supported by evidence and state of the art opinions in my judgment. Moreover, in the introduction the authors state that "but the crosstalk among adjacent pixels will significantly affect its spatial resolution and practical applications 9-16".

However, in my opinion, references 9 to 16 are not supporting such sentence.

I would suggest further smoothing the narrative through the whole paper (not just in these parts that I am mentioning here): you may claim the development of a sensor without cross-talk, but please to not associate this to biomimetism or to a strict technological requirement. You can as an example associate your design choice to ease of sensor calibration, but other solutions (with cross-talk) could show other benefits (redundancy, resiliency to disturbance, fault tolerance, ...).

Please consider my comments as constructive notes to contribute improving the paper, which has added value in my opinion and deserves attention for publication as soon as these aspects will be properly framed.

Response:

We would like to express our sincere thanks to the referee for your great effort to review the manuscript and positive evaluation on our work. After your guidance and my review of relevant literature (Sci. Robot. 2022, 7, eabm0608 and Nat. Mach. Intell. 2022, 4, 425-+), crosstalk can indeed amplify and locate the external stimuli, but in order to realize the accurate tactile perception, data analysis or machine learning will be needed, which will increase the complexity of the whole system. For flexible electronics, if the measured signal is not calibrated and analyzed, and the sensor array shows poor isolation ability for unnecessary crosstalk, the imaging boundary will be fuzzy, which is not conducive to high-precision detection. From this perspective, we need to optimize the device structure to isolate different pixels, so a strain localization films (*prsl*/PDMS) is introduced to alleviate the mechanical crosstalk. This method is expected to achieve good tactile perception in high-density sensor array, while also reducing the dependence

on back-end data processing system. Based on the above analysis, we have revised the paper to reduce the emphasis on crosstalk isolation, and further demonstrated the advantages of the sensor structure for improving detection accuracy.

The main text and the Supplementary Note are revised according to your comment, which is listed as following for your convenience (all changes made in the revised MS is highlighted in yellow):

Main text, Abstract, Page 2

Tactile sensors with high spatial resolution are crucial to manufacture large scale flexible electronics, and low crosstalk sensor array combined with advanced data analysis is beneficial to improve detection accuracy. Here, we demonstrated the photoreticulated strain localization films (*prsl*/PDMS) to prepare the ultralow crosstalk sensor array, which form a micro-cage structure to reduce the pixel deformation overflow by 90.3% compared to that of conventional flexible electronics. It is worth noting that *prsl*/PDMS acts as an adhesion layer and provide spacer for pressure sensing. Hence, the sensor achieves the sufficient pressure resolution to detect 1g weight even in bending condition, and it could monitor human pulse under different states or analyze the grasping postures. Experiments show that the sensor array acquires clear pressure imaging and ultralow crosstalk (33.41 dB) without complicated data processing, indicating that it has a broad application prospect in precise tactile detection.

Main text, Page 3

Development of large-scale and high-density flexible sensor arrays could provide better human-machine interaction¹⁻⁸, and high-precision tactile perception could be achieved by analyzing the measured data^{9, 10} or isolating various crosstalk¹¹⁻¹⁶. At present, signal crosstalk in sensor arrays is mainly divided into two types. One is the crosstalk among electrical signals, such as leakage, breakdown or the external electromagnetic interference, etc. This phenomenon could be alleviated by using inductance and capacitance, or solved by the signal processing algorithm. The other type is so-called mechanical crosstalk, which often occurs in flexible electronics. When a pixel receives an external pressure, its deformation will inevitably spread to surrounding regions, so the adjacent pixels will also respond. Crosstalk may be used to localize the external stimulus, but it usually requires further analysis of the measured signal for higher detection accuracy.

Main text, Page 4

Furthermore, Oddo¹⁰ et al reported a biomimetic skin based on the photonic fiber Bragg grating transducers with overlapping receptive fields, and the force and localization predictions could be realized by the convolution neural deep learning algorithm. This technology combined with an efficient calculation power system will show broad application prospects in the field of collaborative robots. As discussed above, proposing an effective strategy to optimize the sensor structure to achieve direct

measurement of external signals can not only greatly reduce in-depth analysis of data, but also effectively achieve precise tactile detection in high spatial resolution sensors.

Main text, Page 13

The cage-structured sensor based on *prsi*/PDMS layer could effectively avoid the diffusion of strain, and clear multi-pixel stimulation could be obtained through direct measurement, greatly reducing the back-end calibration processing.

Main text, Page 20-21

9. Sun Huanbo, Martius Georg. Guiding the design of superresolution tactile skins with taxel value isolines theory. *Sci. Robot.* 7, eabm0608 (2022).

10. Massari Luca, et al. Functional mimicry of Ruffini receptors with fibre Bragg gratings and deep neural networks enables a bio-inspired large-area tactile-sensitive skin. *Nat. Mach. Intell.* 4, 425-+ (2022).